EMBO
Molecular Medicine

# Dysregulated mesenchymal PDGFR-β drives kidney fibrosis

Eva M Buhl[1,2,3], Sonja Djudjaj[1], Barbara M Klinkhammer[1], Katja Ermert[1], Victor G Puelles[2,4,5], Maja T Lindenmeyer[4], Clemens D Cohen[6], Chaoyong He[7,8], Erawan Borkham-Kamphorst[9], Ralf Weiskirchen[9], Bernd Denecke[10], Panuwat Trairatphisan[11], Julio Saez-Rodriguez[11] iD, Tobias B Huber[4], Lorin E Olson[7], Jürgen Floege[2] & Peter Boor[1,2,*] iD

## Abstract

Kidney fibrosis is characterized by expansion and activation of platelet-derived growth factor receptor-β (PDGFR-β)-positive mesenchymal cells. To study the consequences of PDGFR-β activation, we developed a model of primary renal fibrosis using transgenic mice with PDGFR-β activation specifically in renal mesenchymal cells, driving their pathological proliferation and phenotypic switch toward myofibroblasts. This resulted in progressive mesangioproliferative glomerulonephritis, mesangial sclerosis, and interstitial fibrosis with progressive anemia due to loss of erythropoietin production by fibroblasts. Fibrosis induced secondary tubular epithelial injury at later stages, coinciding with microinflammation, and aggravated the progression of hypertensive and obstructive nephropathy. Inhibition of PDGFR activation reversed fibrosis more effectively in the tubulointerstitium compared to glomeruli. Gene expression signatures in mice with PDGFR-β activation resembled those found in patients. In conclusion, PDGFR-β activation alone is sufficient to induce progressive renal fibrosis and failure, mimicking key aspects of chronic kidney disease in humans. Our data provide direct proof that fibrosis per se can drive chronic organ damage and establish a model of primary fibrosis allowing specific studies targeting fibrosis progression and regression.

**Keywords** anemia; chronic kidney disease; fibroblasts; PDGFR; progression
**Subject Category** Urogenital System

See also: **A Ortiz** (March 2020)

## Introduction

Fibrosis is a common pathological endpoint of chronic, progressive, or severe acute diseases in virtually all organs. Chronic kidney diseases, which affect 9–16% of the world population, are a prototypical disease leading to organ fibrosis (Jha *et al*, 2013). Due to organ-overlapping similarities in fibrosis, finding potential treatments for fibrosis is a major research focus, strongly relying on animal models. However, in virtually all models fibrosis develops secondary to another type of injury, hampering and obscuring the understanding of fibrogenesis itself.

Fibrosis centrally involves deposition of extracellular matrix by activated mesenchymal cells (Boor & Floege, 2012). Adult renal mesenchymal cells, which all retain embryonal PDGFR-β expression, include glomerular mesangial cells, cortical fibroblasts, pericytes of the medulla, and vascular smooth muscle cells (Alpers *et al*, 1992). PDGFR-β is a tyrosine-kinase receptor for PDGF-B and PDGF-D. Upon activation, PDGFR-β induces downstream signaling triggering cell proliferation, migration, and differentiation (Tallquist & Kazlauskas, 2004; Floege *et al*, 2008; Boor *et al*, 2014; Demoulin & Essaghir, 2014; Klinkhammer *et al*, 2018). Ubiquitous *Pdgfrb* deletion or deficiency for PDGF-B in mice led to prenatal death due to severe vascular, hematological, and renal defects (Soriano, 1994; Lindahl, 1997; Arar *et al*, 2000). Conversely, increased PDGFR-β signaling resulting from activating mutations, gene translocation, or increased ligand abundance is involved in the pathogenesis of malignancies, atherosclerosis, and organ fibrosis (Bonner, 2004; Andrae *et al*, 2008; Demoulin & Essaghir, 2014; Klinkhammer *et al*, 2018). Activating mutations of PDGFR-β are also found in rare human genetic syndromes (Johnston *et al*,

1   Institute of Pathology, RWTH University of Aachen, Aachen, Germany
2   Division of Nephrology, RWTH University of Aachen, Aachen, Germany
3   Electron Microscopy Facility, RWTH University of Aachen, Aachen, Germany
4   III. Department of Medicine, University Medical Center Hamburg-Eppendorf, Hamburg, Germany
5   Department of Nephrology, Monash Health, and Center for Inflammatory Diseases, Monash University, Melbourne, Vic., Australia
6   Nephrological Center, Medical Clinic and Policlinic IV, University of Munich, Munich, Germany
7   Cardiovascular Biology Program, Oklahoma Medical Research Foundation, Oklahoma City, OK, USA
8   State Key Laboratory of Natural Medicines, Department of Pharmacology, China Pharmaceutical University, Nanjing, China
9   Institute of Molecular Pathobiochemistry, Experimental Gene Therapy and Clinical Chemistry, RWTH University of Aachen, Aachen, Germany
10  Interdisciplinary Center for Clinical Research (IZKF), RWTH University of Aachen, Aachen, Germany
11  Faculty of Medicine, Institute for Computational Biomedicine, Heidelberg University, and Heidelberg University Hospital, Heidelberg, Germany
    *Corresponding author. Tel: +49 241 8085227; Fax: +49 241 8082446; E-mail: pboor@ukaachen.de

2015; Takenouchi *et al*, 2015; Minatogawa *et al*, 2017) and in different malignancies (Cheung *et al*, 2013; Martignetti *et al*, 2013; Johnston *et al*, 2015; Arts *et al*, 2016). However, consequences of increased PDGFR-β activity for kidney development are currently unknown.

The role of renal mesenchymal PDGFR-β in adults is assumed to be minor under physiological conditions (Liao *et al*, 1996). During kidney diseases, PDGFR-β is upregulated, and blocking PDGFR-β signaling using either specific antibodies (Takahashi *et al*, 2005) or non-specific tyrosine-kinase receptor inhibitors such as imatinib ameliorated kidney injury in various renal disease models (Lassila *et al*, 2005; Wang *et al*, 2005; Zoja *et al*, 2006; Floege *et al*, 2008). In addition, neutralization of the PDGFR-β ligands PDGF-B (Floege *et al*, 1999; Ostendorf *et al*, 2001) or PDGF-D (Ostendorf, 2003; Boor *et al*, 2007; Buhl *et al*, 2016) was also beneficial in models of kidney injury.

To more specifically study the consequences of PDGFR-β activation in kidney fibrosis in patients, we developed a murine model with renal mesenchymal cell-specific PDGFR-β activation. The model mimics various aspects of CKD in patients, allows specific analyses of fibrogenesis, and provides first direct evidence that fibrosis itself is harmful and leads to organ injury.

# Results

### PDGFR-β expression in patients with kidney fibrosis

Compared to kidney samples from patients without fibrosis, fibrotic kidneys showed significantly increased expression and phosphorylation of PDGFR-β (Fig 1A). Fibrosis was confirmed by increased expression of α-smooth muscle actin (α-SMA; Fig 1A). PDGFR-β was expressed in renal mesangial cells, interstitial fibroblasts and pericytes, and vascular smooth muscle cells (Fig 1B). In fibrotic and diseased kidneys, independently of the underlying diseases, i.e., diabetic nephropathy, hydronephrosis, interstitial nephritis, polycystic kidney disease, and transplant rejection, PDGFR-β expression was increased. Using optical tissue clearing and 3D reconstruction (Puelles *et al*, 2019) in *Pdgfrb-GFP* reporter mice, we confirmed that PDGFR-β is specifically expressed in renal mesenchymal cells in all kidney compartments and highly upregulated during fibrosis as in patients (Fig 1C and D; Movies EV1 and EV2), suggesting that murine models are relevant to study the effects of PDGFR-β activation.

### Murine model with renal mesenchymal cell-specific PDGFR-β activation

To specifically manipulate PDGFR-β in the renal mesenchyme, we targeted the FoxD1 lineage of cells using *FoxD1-Cre* mice (*FoxD1^GC^*). FoxD1 is expressed in mesenchymal progenitor cells of the metanephric mesenchyme (Hatini *et al*, 1996; Gomez & Duffield, 2014). To visualize cells originating from this population, we crossbred the *Foxd1-Cre* line with tdTomato reporter mice (B6;129S6-*Gt(ROSA)26Sor^tm9(CAG-tdTomato)Hze^*/J). tdTomato⁺ cells were only found in expected mesangial and interstitial locations (Appendix Fig S1). PDGFR-β immunofluorescence staining confirmed a nearly complete overlap of the tdTomato⁺ and PDGFR-β⁺ cells, verifying the

specificity of this approach to target the renal mesenchyme and the origin of renal PDGFR-β⁺ cells from the FoxD1 lineage (Appendix Fig S1).

### PDGFR-β activation in renal mesenchymal cells induces their proliferation

To generate mice with a renal mesenchymal cell-specific, constitutively active PDGFR-β, we used mice in which one wt *Pdgfrb* allele was substituted by a conditional knock-in of *Pdgfrb* with an activating point mutation (V536A) in the juxtamembrane domain of PDGFR-β, denoted as "J" (*Pdgfrb^+/J^*), behind a floxed STOP cassette (Olson & Soriano, 2011). The expression of this receptor mutant occurs via the endogenous *Pdgfrb* promoter and only after excision of the floxed STOP cassette by a Cre recombinase. In the absence of Cre recombinase, these mice only have one active *Pdgfrb* allele, allowing analyses of gene-dose effects. Compared to mice with two wt *Pdgfrb* alleles, hemizygous *Pdgfrb^+/J^* mice did not show any significant reduction in PDGFR-β protein in the kidney, developed normally (Appendix Fig S2A–C), and showed similar development of fibrosis in a model of renal interstitial fibrosis (i.e., on day 5 after unilateral ureter ligation; Appendix Fig S2D and E). These data suggested that in mice, a single allele of *Pdgfrb* is sufficient for normal kidney but also fibrosis development, showing that these mice can be used as relevant controls similarly to wt mice.

To activate PDGFR-β signaling in renal FoxD1⁺ mesenchymal cells, we generated *Foxd1Cre::Pdgfrb^+/J^* mice (Fig 2A, Appendix Table S1, Fig EV1A and B). These mice indeed showed increased phosphorylation of PDGFR-β in kidney cortex lysates and phosphorylation of the downstream signaling molecules Akt and p38 (Fig EV1C and D). Six- to 35-week-old *Foxd1Cre::Pdgfrb^+/J^* mice exhibited significantly increased proliferation of renal mesenchymal cells in all glomeruli and the whole interstitium compared to wt mice (Fig 2B–D). In contrast, proliferation of tubular epithelial cells did not differ between the two groups (Fig 2E). We next crossbred the mice with a red-fluorescence tdTomato reporter, which is activated by Cre recombinase and allowed us to trace cells of FoxD1 lineage. Compared to *Foxd1Cre::tdTomato* mice, *Foxd1Cre::Pdgfrb^+/J^::tdTomato* mice showed a prominent increase in the number of red-fluorescent glomerular mesangial cells as well as cortical interstitial fibroblasts (Fig 2F and G). Consistent with our *in vivo* data, *in vitro*, primary renal fibroblasts and mesangial cells isolated from *Foxd1Cre::Pdgfrb^+/J^* mice showed a 2.2- and 2.7-fold increase in proliferation compared to cells from wt littermates (Fig 2H).

### Renal mesenchymal PDGFR-β activation induces progressive kidney fibrosis

Starting at 6 weeks and progressively increasing with age, glomeruli of *Foxd1Cre::Pdgfrb^+/J^* mice exhibited diffuse and global mesangioproliferative glomerulonephritis with increasing glomerular tuft size (Fig 3A and B). In addition, mesangial cells exhibited a switch toward a profibrotic myofibroblast-like phenotype, characterized by *de novo* expression of the myofibroblast marker α-SMA (Fig 3C and D). Starting at week 14, there was also increasing deposition of extracellular matrix in the mesangium, i.e., mesangial sclerosis (Fig 3E and F, Appendix Fig S3). Deeper glomeruli, i.e., ontogenetically oldest glomeruli, showed much more prominent changes

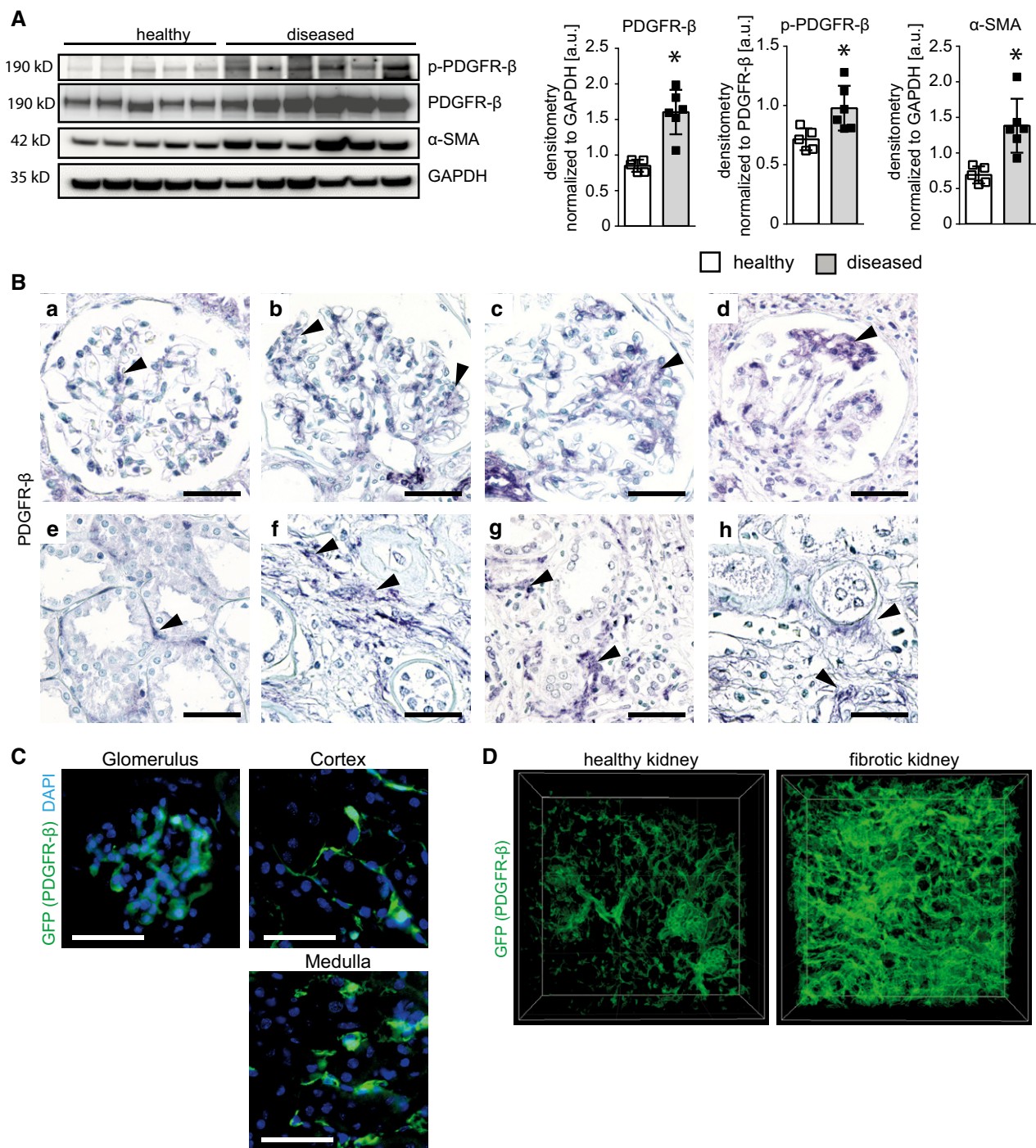

**Figure 1. PDGFR-β expression is upregulated in fibrotic human and murine kidneys.**

A Immunoblot detection and its quantification of PDGFR-β in kidney cortex lysates of healthy and diseased human kidneys show twice as much PDGFR-β abundance in diseased tissue. A stronger signal for phosphorylated PDGFR-β indicates the active status of the receptor. The high α-SMA content in the diseased kidneys shows that they are affected by fibrosis. Bar graphs show means ± SD, healthy $n = 5$, diseased $n = 6$. Statistical analysis was performed by unpaired two-tailed Student's *t*-test. *$P < 0.05$. Exact *P*-values are provided in Appendix Table S4.

B PDGFR-β staining is detectable in mesangium of healthy (a) and diseased (b–d) glomeruli of human kidneys. PDGFR-β-positive mesangial cells expand in mesangioproliferating and scarring glomerular changes. Similarly, the population of PDGFR-β-positive interstitial cells is discrete in healthy kidney cortex (e) but expands in fibrosis (f–h). Arrows point to selected areas of positive PDGFR-β staining. Scale bar = 50 μm.

C GFP expression under *Pdgfrb* promoter in mice is present in mesangial cells of glomeruli and interstitial cells of cortex and medulla, but not in tubular cells. Nuclei are stained with DAPI (blue). Scale bar = 50 μm.

D Tissue clearing with *Scale* and 3D reconstruction of *Pdgfrb-GFP* reporter mice in a healthy and fibrotic kidney (UUO day 5) shows the expansion of *Pdgfrb*-expressing cells in fibrosis.

Source data are available online for this figure.

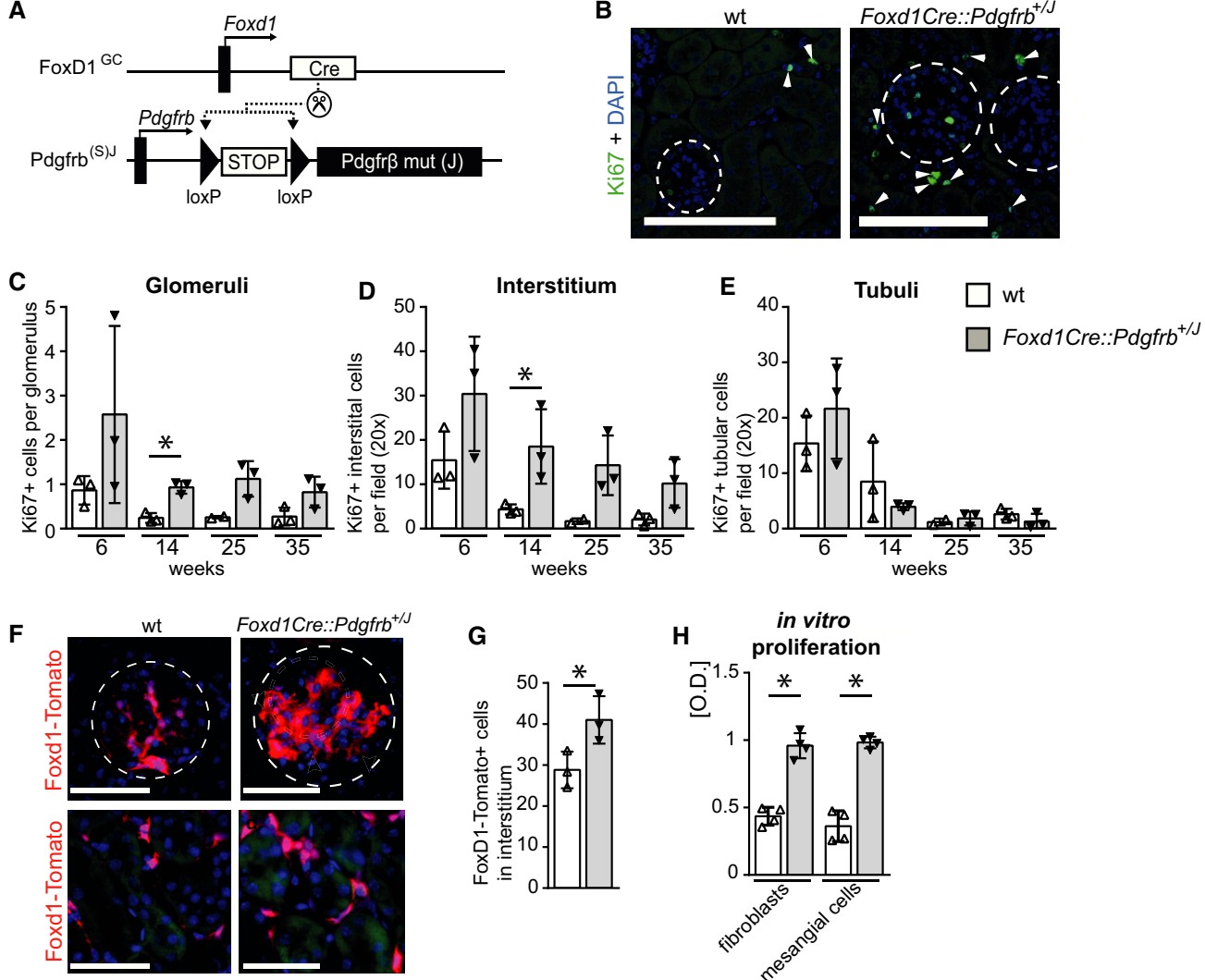

**Figure 2. PDGFR-β activation leads to mesenchymal proliferation *in vivo* and in *vitro*.**

A   Activation of PDGFR-β in renal mesenchymal cells, denoted here as *Foxd1Cre::Pdgfrb*[+/J] mice, was achieved by crossbreeding the *Foxd1-Cre* mouse line (*FoxD1*[GC]) with a heterozygous knock-in mouse line with constitutively active *Pdgfrb* mutant (J) allele (*Pdgfrb*[(S)J]) instead of the wt *Pdgfrb* allele.

B   Representative Ki67 immunofluorescence staining (green) in wt and *Foxd1Cre::Pdgfrb*[+/J] mice, showing increased proliferation in the transgenic mice. Glomeruli are outlined with circles, and arrowheads point to Ki67-positive interstitial cells. Nuclei are stained with DAPI (blue). Scale bar = 50 μm.

C–E   Quantification of proliferating Ki67-positive cells specifically in glomeruli (C), interstitium (D), and tubules (E) in *Foxd1Cre::Pdgfrb*[+/J] mice (black bars) and wt mice (white bars) 6, 14, 25, and 35 weeks of age. *Foxd1Cre::Pdgfrb*[+/J] mice exhibited increased proliferation of glomerular and interstitial cells, whereas tubular epithelial cell proliferation was not altered. Data in (C–E) are shown as means ± SD of n = 3 animals per group. Statistical analysis was performed by unpaired two-tailed Student's *t*-test. *P < 0.05 compared to wt of the same time point. Exact P-values are provided in Appendix Table S4.

F   FoxD1 reporter mice (*Foxd1::tdTomato*) confirmed the expansion of mesenchymal cells in 14-week-old *Foxd1Cre::Pdgfrb*[+/J] mice in both glomeruli and interstitium. Circles outline glomeruli. Scale bar = 50 μm.

G   Quantification of FoxD1-Tomato-positive cells in the cortical interstitium in 14-week-old *Foxd1Cre::Pdgfrb*[+/J] and wt mice confirmed the significantly increased expansion of mesenchymal cells by 42% in *Foxd1Cre::Pdgfrb*[+/J] mice. Cells were counted in six view fields at 40× magnification. Bar graphs show means ± SD of n = 3 mice per group. Statistical analysis was performed by unpaired two-tailed Student's *t*-test. *P < 0.05 compared to wt of the same time point. Exact P-values are provided in Appendix Table S4.

H   Isolated primary fibroblasts and mesangial cells from *Foxd1Cre::Pdgfrb*[+/J] mice have higher proliferation rates assessed by bromodeoxyuridine (BrdU) incorporation assay compared to cells from wt mice. Bar graphs show means ± SD, n = 4 per group, Statistical analysis was performed by unpaired two-tailed Student's *t*-test. *P < 0.05. Exact P-values are provided in Appendix Table S4.

Source data are available online for this figure.

compared to more superficial glomeruli. All this occurred in the virtual absence of glomerular immune cell infiltration (Appendix Fig S3).

Transmission electron microscopy confirmed prominent expansion of the mesangium without electron-dense deposits excluding complement or immunoglobulin deposition. Glomerular endothelial cells, the

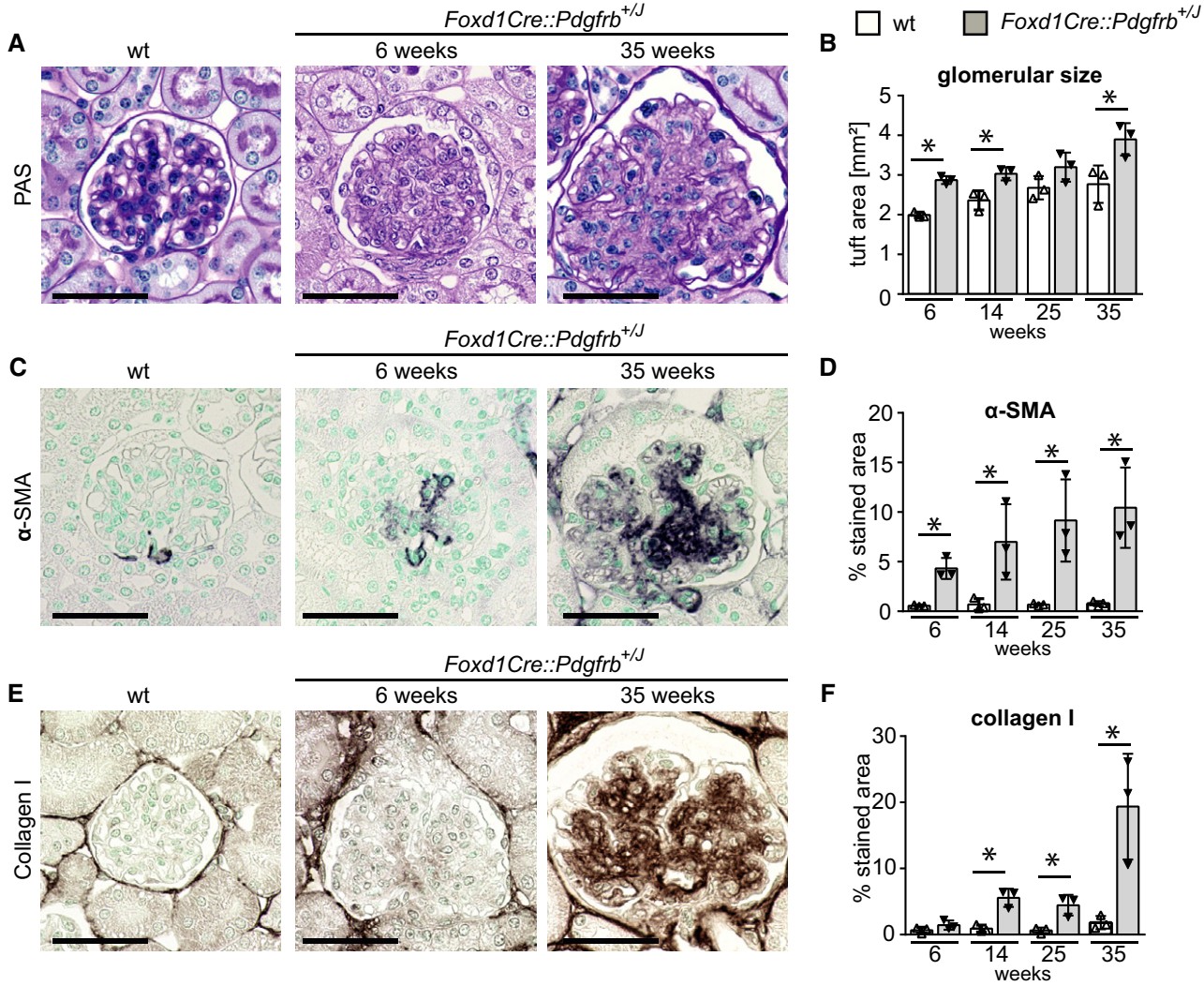

**Figure 3. Activation of PDGFR-β in mesenchymal cells in *Foxd1Cre::Pdgfrb*[+/J] mice results in progressive mesangioproliferative glomerulonephritis and mesangial glomerulosclerosis.**

A  PAS staining of glomeruli of 25-week-old wt and of 6- and 35-week-old *Foxd1Cre::Pdgfrb*[+/J] mice. Glomeruli of *Foxd1Cre::Pdgfrb*[+/J] mice show pathological changes.

B  Glomerular tuft size of *Foxd1Cre::Pdgfrb*[+/J] mice was significantly increased compared to wt mice during the time course.

C, D  (C) Immunohistological staining of α-SMA and its histomorphometric quantification (D) reveal expansion of activated mesangial cells in the glomerular tuft of the transgenic mice in the time course.

E, F  (E) Immunohistological staining of collagen I and its histomorphometric quantification (F) show that collagen I deposits in an increasing manner in *Foxd1Cre::Pdgfrb*[+/J] mice during the time course.

Data information: Scale bars = 50 μm. Bar graphs show mean ± SD of n = 3 animals per group. Statistical analysis was performed by unpaired two-tailed Student's *t*-test. *P < 0.05 compared to wt of the same time point. Exact P-values are provided in Appendix Table S4.
Source data are available online for this figure.

podocytes, and the peripheral glomerular basement membranes appeared normal (Appendix Fig S4). In line with this, we found no proteinuria or hematuria in these mice (tested via urine test strips).

In the tubulointerstitium of *Foxd1Cre::Pdgfrb*[+/J] mice, the number of fibroblasts increased steadily with age (Fig 4A and B). In older mice, some of these cells showed *de novo* expression of α-SMA (Fig 4A). Increased deposition of extracellular matrix was also found in the *Foxd1Cre::Pdgfrb*[+/J] mice (Fig 4C and D, Appendix Fig S5). In contrast to the progressive changes in glomeruli, tubulointerstitial fibrosis was already prominently increased in young mice in cortical and medullary regions

(Appendix Fig S6). Fibrosis and expansion of fibroblasts were confirmed by transmission electron microscopy (Appendix Fig S7), which also demonstrated fibroblasts with a prominent dilation of the endoplasmatic reticulum, supporting increased protein synthesis, collagen processing (Canty-Laird *et al*, 2012), and cell stress (Canty-Laird *et al*, 2012; Schonthal, 2012). Endothelial cells of the peritubular capillaries showed thickening and loss of fenestrations, all common in renal fibrosis (Babickova *et al*, 2017). Thus, mesangial cells appear to provide little support for the glomerular endothelium, whereas interstitial fibroblasts and pericytes are important for maintaining peritubular capillary architecture.

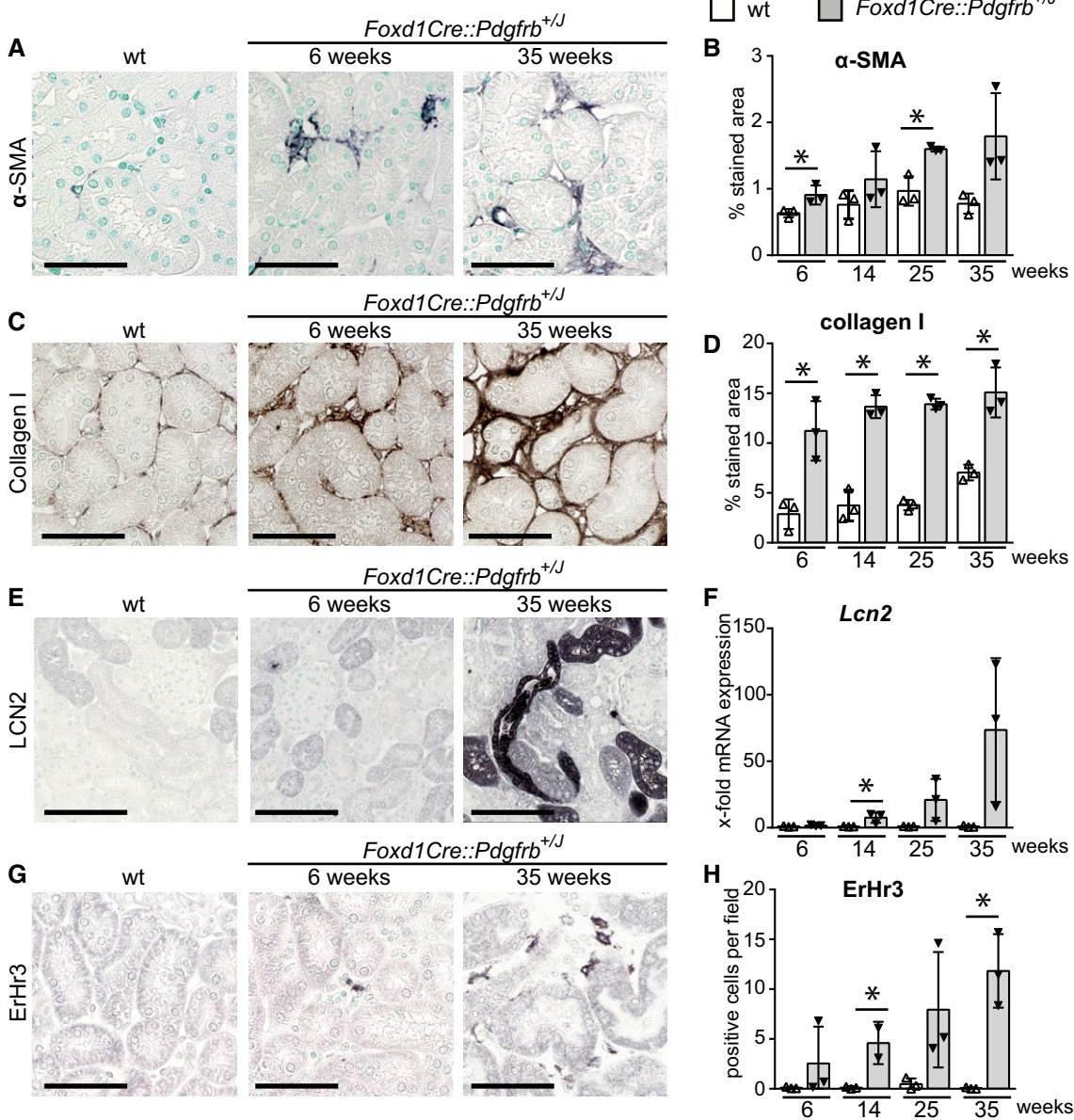

**Figure 4. Activation of PDGFR-β in mesenchymal cells in *Foxd1Cre::Pdgfrb*^+/J^ mice results in progressive interstitial fibrosis, inflammation, and tubular damage.**

A, B (A) Histological stainings and its histomorphometric quantification (B) of kidney cortex for α-SMA of wt (25 weeks) and *Foxd1Cre::Pdgfrb*^+/J^ mice 6 and 35 weeks of age show expansion of myofibroblasts in the interstitium of the transgenic mice.

C, D (C) Collagen I staining and its histomorphometric quantification (D) show the progression of interstitial fibrosis in the time course.

E, F The tubular stress marker lipocalin-2 (LCN2) is not expressed in early time points but increases in the time course in late stages when fibrosis is more prominent, as seen on histological staining (E) and mRNA level (F).

G, H A slight, but significant, increase in interstitial ErHr3-positive stained macrophages was observed, particularly at later stages (G). For evaluation, ErHr3-positive cells were counted on 20 view fields on 40× magnification (H).

Data information: Scale bars = 50 μm. Bar graphs show means ± SD of *n* = 3 animals per group. Statistical analysis was performed by unpaired two-tailed Student's *t*-test. *$P < 0.05$ compared to wt of the same time point. Exact *P*-values are provided in Appendix Table S4.

Source data are available online for this figure.

At 6 weeks of age, all tubules of the *Foxd1Cre::Pdgfrb*^+/J^ animals appeared morphologically normal and failed to express the injury marker lipocalin-2 (LCN2). Starting at week 14, whole tubular segments stained strongly positive for LCN2, paralleled by an up to 73.5-fold increased expression of *Lcn2* mRNA (Fig 4E and F). Quantification of inflammatory infiltrates showed a minor increase in the number of ErHr3-positive interstitial macrophages (Fig 4G and H) as well as CD45- and F4/80-positive inflammatory cells at 25 and 35 weeks of age (Appendix Fig S5).

The tdTomato reporter mice showed a population of *Foxd1*-derived mesenchymal cells underneath the renal capsule (Appendix Fig S8), in line with a previous report (Levinson *et al*, 2005). These cells expanded and became profibrotic in the *Foxd1Cre::Pdgfrb*$^{+/J}$ mice, resulting in prominent fibrotic thickening of the renal capsule.

### Renal mesenchymal PDGFR-β activation induces kidney failure and anemia but no hypertension

Aging *Foxd1Cre::Pdgfrb*$^{+/J}$ mice exhibited progressive kidney failure with a decline in creatinine clearance (Fig 5A) and an increase in blood urea nitrogen (BUN) (Fig 5B), which exceeded that observed in aging wt mice.

Renal fibroblasts are the major source of erythropoietin (EPO). When fibroblasts convert to a myofibroblast phenotype in models of renal fibrosis, they lose the ability to produce EPO (Souma *et al*, 2013). In *Foxd1Cre::Pdgfrb*$^{+/J}$ mice aged 6 weeks, erythrocyte numbers (Fig 6A), hematocrit (Fig 6B), and hemoglobin (Fig 6C) were similar between wt and *Foxd1Cre::Pdgfrb*$^{+/J}$ mice. However, from week 14 onward progressive anemia developed (Fig 6A). Similar to patients with chronic kidney disease, EPO serum levels failed to increase in the anemic *Foxd1Cre::Pdgfrb*$^{+/J}$ mice (Fig 6D). *In situ* hybridization for *Epo* mRNA confirmed a reduced number of

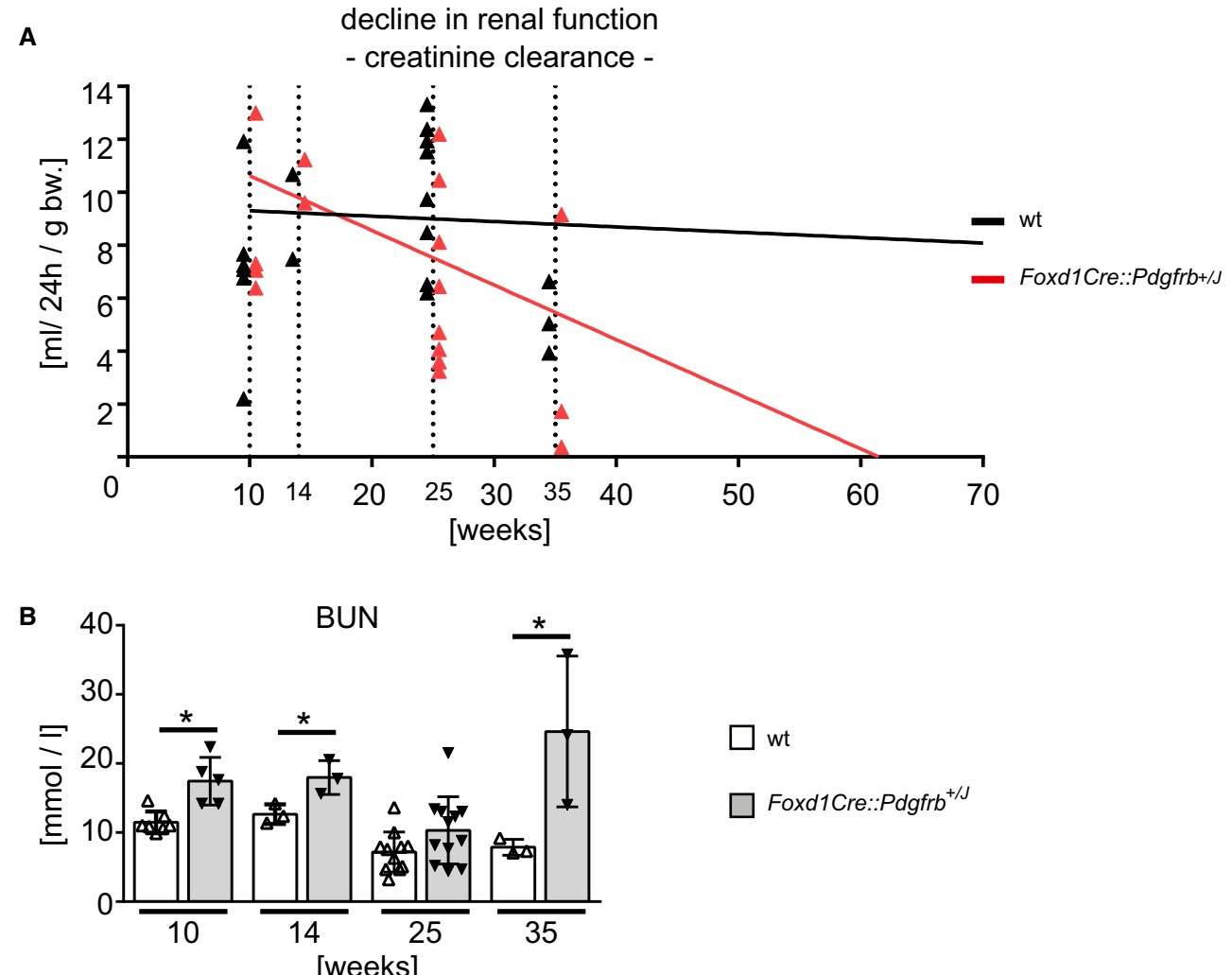

**Figure 5. Reduced kidney function in *Foxd1Cre::Pdgfrb*$^{+/J}$ mice.**

A Creatinine clearance was measured in mice 10, 14, 25, and 35 weeks of age and decreased continuously in *Foxd1Cre::Pdgfrb*$^{+/J}$ mice, with a predicted complete loss of renal function 62 weeks in transgenic mice using linear regression analyses as described previously (Steiger *et al*, 2018). Ten weeks n = 5 wt and n = 4 *Foxd1Cre::Pdgfrb*$^{+/J}$, 14 weeks n = 2 per group, 25 weeks n = 8 wt and n = 8 *Foxd1Cre::Pdgfrb*$^{+/J}$, 35 weeks n = 3 per group.

B The blood urea nitrogen (BUN) concentrations were higher in *Foxd1Cre::Pdgfrb*$^{+/J}$ mice compared to wt mice, apart from the 25-week-old group. Bar graphs show means ± SD; 10 weeks n = 7 wt and n = 5 *Foxd1Cre::Pdgfrb*$^{+/J}$, 14 weeks n = 3 per group, 25 weeks n = 11 wt and n = 12 *Foxd1Cre::Pdgfrb*$^{+/J}$, 35 weeks n = 3 per group. Statistical analysis was performed by unpaired two-tailed Student's *t*-test. *P < 0.05 compared to wt of the same time point. Exact *P*-values are provided in Appendix Table S4.

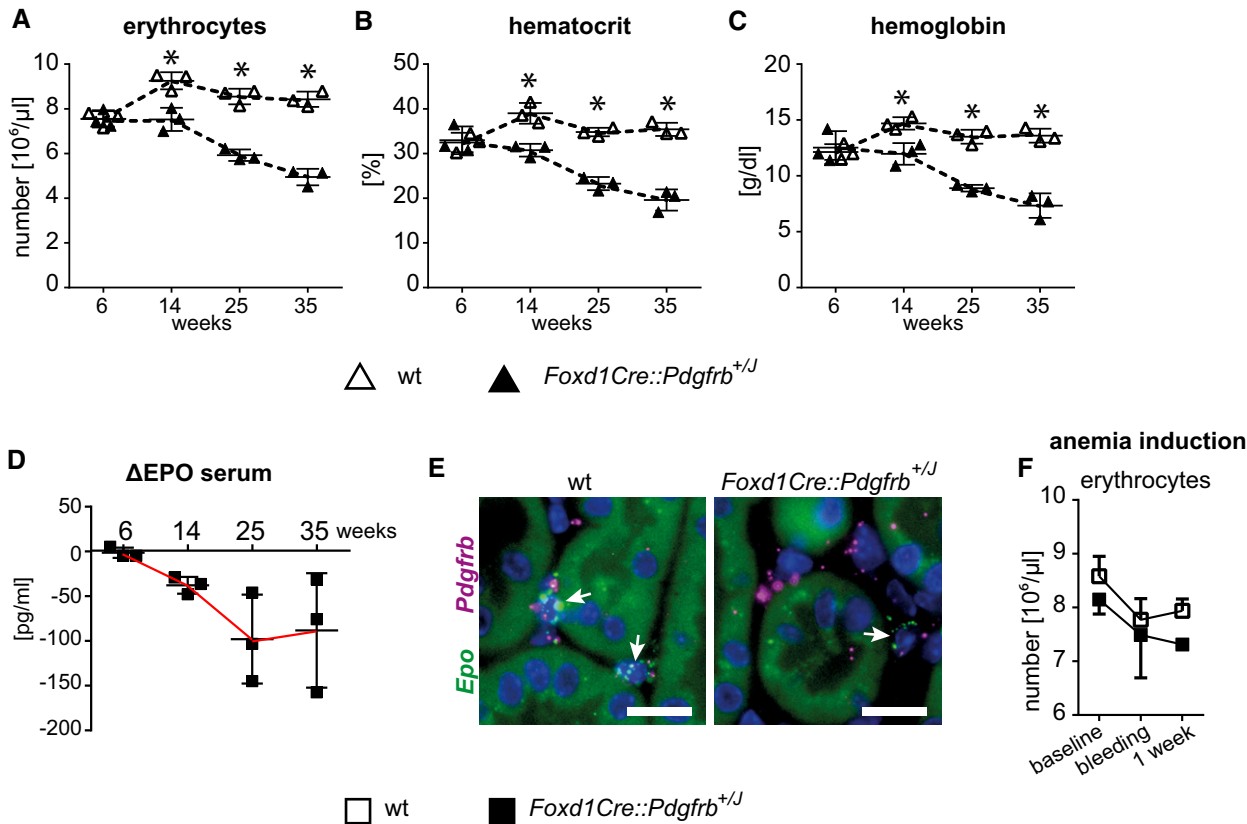

**Figure 6.** The *Foxd1Cre::Pdgfrb*⁺/ᴶ mice develop progressive renal anemia and are unable to regulate erythropoiesis under pathological stimuli.

A–D Erythrocyte numbers (A), hematocrit (B), and hemoglobin content (C) decrease in *Foxd1Cre::Pdgfrb*⁺/ᴶ mice continuously. (D) EPO levels shown as delta from respective wt littermates. Horizontal lines are means ± SD of n = 3 per group. Statistical analysis was performed by unpaired two-tailed Student's *t*-test. *P < 0.05 compared to wt of the same time point. Exact *P*-values are provided in Appendix Table S4.

E RNA *in situ* hybridization (RNAscope) of *Epo* (green) and *Pdgfrb* (purple) in renal kidney cortex of 35-week-old mice shows that EPO-producing cells co-express *Pdgfrb*. *Foxd1Cre::Pdgfrb*⁺/ᴶ mice have less EPO-producing cells (arrows) than wt mice. Scale bar = 25 μm.

F After induction of anemia by blood taking, the erythrocyte numbers recovered after 1 week in wt mice, whereas *Foxd1Cre::Pdgfrb*⁺/ᴶ mice were not able to recover their erythrocytes numbers at all. Data are means ± SD of n = 5 per group. Statistical analysis was performed by unpaired two-tailed Student's *t*-test. *P < 0.05 compared to wt of the same time point. Exact *P*-values are provided in Appendix Table S4.

Source data are available online for this figure.

interstitial *Epo*-producing fibroblasts in *Foxd1Cre::Pdgfrb*⁺/ᴶ compared to wt mouse kidneys and showed that these cells express *Pdgfrb,* consistent with their mesenchymal origin (Fig 6E). While blood loss in wt mice resulted in physiological compensatory erythropoiesis, this response was completely absent in *Foxd1Cre:: Pdgfrb*⁺/ᴶ mice (Fig 6F). Thus, activated PDGFR-β in fibroblasts appears to abrogate their ability to produce and regulate EPO, suggesting a novel potential pathway involved in EPO production.

Renin, an important hormone regulating blood pressure, is produced by a small population of specialized vascular smooth muscle of the afferent arterioles, which originate from the FoxD1 lineage (Kurt & Kurtz, 2015). Despite this, the *Foxd1Cre::Pdgfrb*⁺/ᴶ and wt mice remained normotensive at all time points and the number of renin-producing cells remained stable (Fig EV2). These data support a previous report, suggesting that renin production and the pool of renin-producing cells are independent of PDGFR-β signaling (Neubauer *et al*, 2013). These data also showed that our mice represent one of the very few normotensive models of renal fibrosis.

**Renal mesenchymal PDGFR-β activation aggravates progression of renal diseases**

We next challenged the *Foxd1Cre::Pdgfrb*⁺/ᴶ mice with different models of kidney diseases targeting specific compartments, i.e., the unilateral ureteral obstruction (UUO) (Fig 7A and B), which induces a primary interstitial fibrosis without affecting the glomeruli, and angiotensin-2-infusion-induced hypertensive nephropathy (Fig 7C–H), which mainly affects the glomeruli.

In the UUO model, *Foxd1Cre::Pdgfrb*⁺/ᴶ mice developed significantly more renal fibrosis compared to the wt littermates, i.e., 68% higher collagen I deposition and 40% more α-SMA area. F4/80-positive cell infiltrates increased by 127% (Fig 7A and B).

Continuous angiotensin II infusion induced similar hypertension in wt and *Foxd1Cre::Pdgfrb*⁺/ᴶ mice (Fig 7C). After 28 days, wt animals exhibited a 43% increase in BUN but no change in creatinine clearance (Fig 7D and E). Proteinuria and hematuria were absent, and only focal and very mild pathology was observed. In

Figure 7.

**Figure 7. PDGFR-β activation aggravates the course of renal disease models, whereas blocking partially reverses the fibrotic phenotype.**

A, B   Five days after the induction of unilateral ureteral obstruction (UUO), a model of obstructive nephropathy and interstitial fibrosis, we assessed fibrosis (collagen I, α-SMA) and inflammation (F4/80), the classical readout parameters in this model. Compared to wt mice with UUO, *Foxd1Cre::Pdgfrb$^{+/J}$* mice developed significantly stronger fibrosis and inflammation (A). Representative pictures of the histological stainings are shown in (B). Scale bar = 50 μm. All data in (A) and (B) show means ± SD; wt *n* = 4, *Foxd1Cre::Pdgfrb$^{+/J}$ n* = 3. Statistical analysis was performed by unpaired two-tailed Student's *t*-test. *P < 0.05. Exact *P*-values are provided in Appendix Table S4.

C–H   Angiotensin II infusion via an osmotic pump induced hypertension similarly in wt and *Foxd1Cre::Pdgfrb$^{+/J}$* mice (C). In this model of hypertensive injury, the kidney function and even the histopathology were largely unaffected in wt mice, whereas the *Foxd1Cre::Pdgfrb$^{+/J}$* mice showed significantly increased BUN values and prominent glomerulopathy after 28 days compared to the starting values (day 0), to wt animals after 28 days, and to age-matched *Foxd1Cre::Pdgfrb$^{+/J}$* mice without angiotensin II infusion (D). Consistently, the creatinine clearance per g bodyweight decreased in the hypertensive *Foxd1Cre::Pdgfrb$^{+/J}$* mice (E). (F) PAS staining showed more prominent mesangial expansion and intratubular protein casts indicative of proteinuria only in hypertensive *Foxd1Cre::Pdgfrb$^{+/J}$* mice. (G) Hypertension did not induce mesangial expansion and activation in wt mice, but led to a significant increase in hypertensive *Foxd1Cre::Pdgfrb$^{+/J}$* mice as visualized by α-SMA staining. (H) Histomorphometric analysis of α-SMA-positive percentage of the glomerular tuft area. All data in (C–H) show means ± SD, wt *n* = 7, *Foxd1Cre::Pdgfrb$^{+/J}$ n* = 5. Statistical analysis was performed by unpaired two-tailed Student's *t*-test. *P ≤ 0.05 compared to wt of the same time point, §P ≤ 0.05 compared to d0, #P ≤ 0.05 compared to same-aged non-hypertensive mice. Exact *P*-values are provided in Appendix Table S4.

I–P   Twenty-two-week-old *Foxd1Cre::Pdgfrb$^{+/J}$* mice were treated with imatinib (daily gavage, 50 mg/kg bodyweight for 21 days; *Foxd1Cre::Pdgfrb$^{+/J}$* + imatinib) and were compared to control *Foxd1Cre::Pdgfrb$^{+/J}$* mice receiving vehicle (water; *Foxd1Cre::Pdgfrb$^{+/J}$* + water). Age-matched wt mice are indicated as dashed lines. (I) Histomorphometric quantifications showed a significant reduction of α-SMA abundance in the glomerular tufts of imatinib-treated *Foxd1Cre::Pdgfrb$^{+/J}$* mice. (J) Glomerular cellularity evaluated by counting the total number of cells per glomerular tuft, normalized to the area, was significantly reduced in imatinib-treated *Foxd1Cre::Pdgfrb$^{+/J}$* mice compared to control mice. (K, L) Histomorphometric quantifications of collagen I showed no effect of imatinib treatment on mesangial sclerosis in the glomeruli (K), whereas reduced interstitial fibrosis in imatinib-treated *Foxd1Cre::Pdgfrb$^{+/J}$* mice compared to controls was found (L). (M) Hemoglobin serum levels were significantly improved in imatinib-treated mice. (N) Representative pictures of the histological stainings of α-SMA (left) and collagen I (middle: glomeruli; right: cortical interstitium). Scale bar = 50 μm. (O, P) Total PDGFR-β protein levels were decreased in *Foxd1Cre::Pdgfrb$^{+/J}$* mice after imatinib treatment, as shown using Western blot (O) and its densitometric evaluation (P). Bar graphs show means ± SD, *n* = 5 per group. Statistical analysis was performed by unpaired two-tailed Student's *t*-test. *P ≤ 0.05 *Foxd1Cre::Pdgfrb$^{+/J}$* + water versus *Foxd1Cre::Pdgfrb$^{+/J}$* + imatinib.

Source data are available online for this figure.

comparison, *Foxd1Cre::Pdgfrb$^{+/J}$* mice showed a more prominent increase in BUN (Fig 7D) and creatinine clearance decreased by more than a half (Fig 7E) paralleled by prominent glomerular and secondary tubulointerstitial injury (Fig 7F). In the glomeruli, this was characterized with an even more increased expansion of α-SMA+ mesangial cells in hypertensive *Foxd1Cre::Pdgfrb$^{+/J}$* mice compared to the normotensive *Foxd1Cre::Pdgfrb$^{+/J}$* mice (Fig 7G and H). All analyzed parameters in *Foxd1Cre::Pdgfrb$^{+/J}$* mice with hypertensive nephropathy were significantly aggravated compared to unchallenged *Foxd1Cre::Pdgfrb$^{+/J}$* mice.

Taken together, PDGFR-β activation in renal mesenchymal cells aggravates the course of interstitial and glomerular diseases.

**Reversibility of the pathological phenotype by inhibition of PDGFR-β activation**

Next, we treated *Foxd1Cre::Pdgfrb$^{+/J}$* mice for 3 weeks with imatinib, a tyrosine-kinase inhibitor targeting PDGFR-β. To mimic a clinically relevant scenario, we started the treatment in an advanced stage of renal fibrosis, i.e., in 22-week-old mice. Imatinib reduced α-SMA-positive cells in glomeruli compared to the vehicle-treated mice (Fig 7I and N) and normalized glomerular cellularity (Fig 7J). However, mesangial sclerosis, i.e., collagen I-positive area, did not decrease (Fig 7K and N), whereas the interstitial collagen I-positive area was significantly reduced by imatinib (Fig 7L and N). Consistent with the histopathological findings, imatinib also normalized anemia (Fig 7M). The total amount of PDGFR-β also decreased, as did the number of mesenchymal PDGFR-β$^+$ cells (Fig 7O and P).

**Downstream effects of PDGFR-β activation**

To analyze the pathways involved in renal fibrosis development, we performed a gene array of the kidney cortex in 6-week-old *Foxd1Cre::Pdgfrb$^{+/J}$* mice versus wt littermates. At this early time point, there were no secondary effects of fibrosis like tubular cell injury or (micro)inflammation, which could complicate the interpretation of the array data (Fig 8A). Forty-six genes differed more than 1.5-fold between the two groups, including genes involved in proliferation (*Ifi44l, Ifi44, Irf9, Cgref1*), extracellular matrix turnover (*Chil4, Chil3*), apoptosis (e.g., *Ifit1, C3, Casp1, Xaf1, Ctse*), autophagy (*Cd38, Ctse, Trim30d, Trim5, Trim12a, Trim12c*), and inflammation (e.g., *Pydc3, Rtp4, Ccl5, Rac2, Pyhin1, Irgm1*). Nineteen of the 46 upregulated genes are interferon-regulated (e.g., *Ifit1, Ifi44, Xaf1, Eif2aka, PD-L1, Oas1b, Irf8, Irf9, Id1*). We confirmed the upregulation of *Ifit1, Irf8, Casp1*, and *Trim30d* in transgenic mice by real-time PCR (Appendix Fig S9).

To further characterize in which kidney cells these genes are regulated, we analyzed single-cell RNA sequencing data from the *Kidney Interactive Transcriptomics* library (KIT, http://humphre yslab.com/SingleCell). In a data set of murine fibrotic kidneys (14 days after UUO), *Islr (Meflin), Hmcn1, Ms4a6c, Apob, Trim5, Trim12a, Ctse, Racs2, Bank1, Irf9, Eif2ak2, and Id3* expression could be allocated to mesenchymal cells, especially to activated fibroblasts (Fig EV3A). Confirmatory, *Pdgfrb* expression was also detected in these cells. In a data set of human renal graft rejection, *ISLR (MEFLIN), HMCN1, APOB, ID3, C3, F3, EIF2AK2*, and also *PDGFRB* were overexpressed by (myo)fibroblasts or pericytes (Fig EV3B). These data further support our notion of a specific gene signature for PDGFR-β activation in fibrosis across species.

Next, we analyzed the PDGFR-β-induced gene pattern in patients with kidney diseases and fibrosis separately in microdissected glomeruli and tubulointerstitium. Compared to the gene expression profile of the *Foxd1Cre::Pdgfrb$^{+/J}$* mice, patients with hypertensive diabetic and hypertensive nephropathies showed a similar and overlapping upregulation of 60.7 and 53.6% genes in glomeruli, and 85.7 and 64.2% of genes in the tubulointerstitium, respectively (Fig 8A).

Using computational approaches, we next analyzed the putative pathways involved in the *Foxd1Cre::Pdgfrb$^{+/J}$* mice. We used

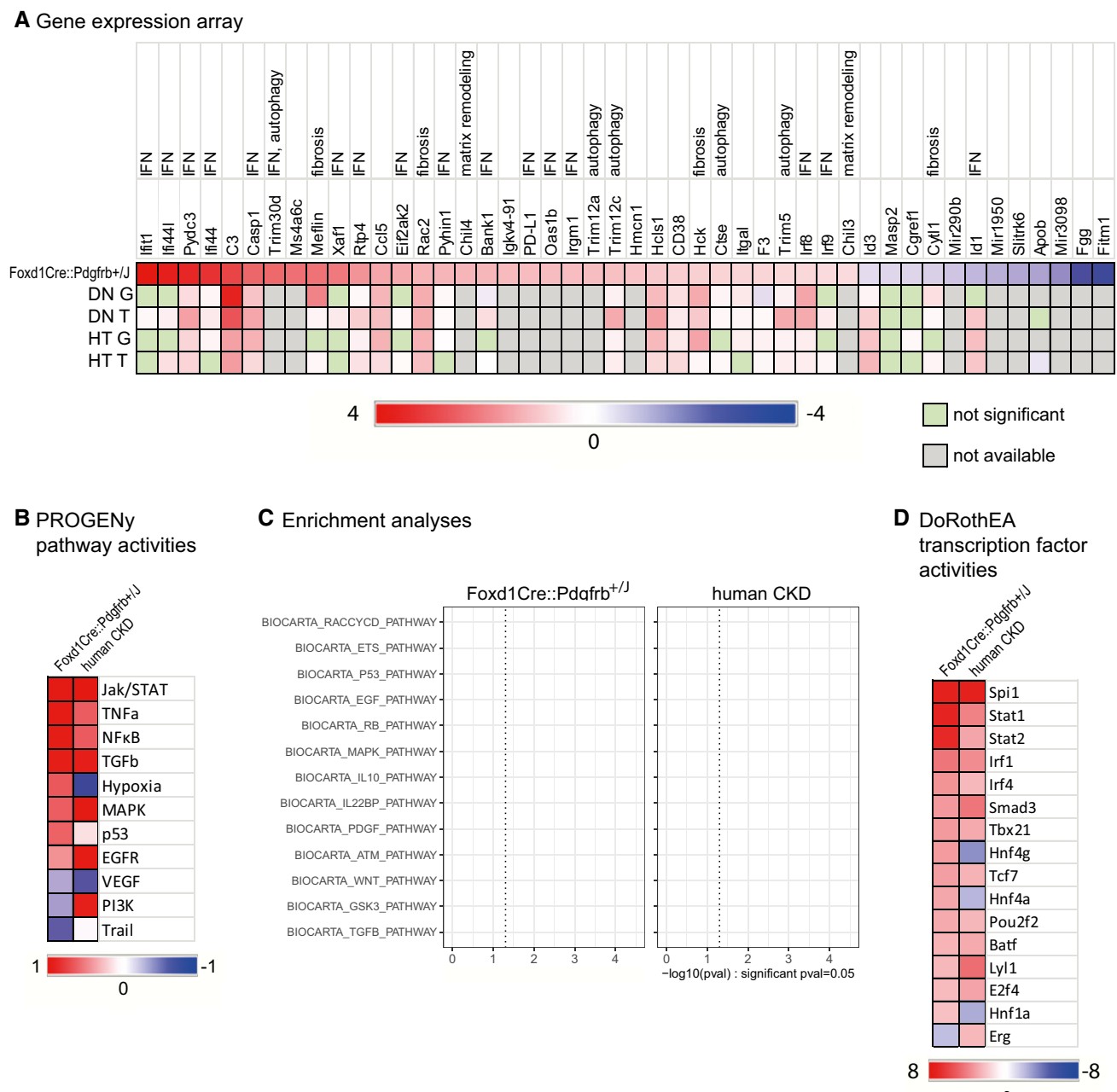

**Figure 8. Foxd1Cre::Pdgfrb+/J mice show similar gene expression profiles like human patients with diabetic or hypertensive nephropathy.**

A   Gene expression arrays were performed in 6-week-old *Foxd1Cre::Pdgfrb+/J* mice compared to wt mice. Listed are genes with log₂ fold changes higher or lower than 0.6. Out of these 46 genes, 17 are IFN-regulated genes, five are involved in autophagy, and five are involved in matrix remodeling and fibrosis. The same genes were analyzed in microdissected glomeruli (G) and tubulointerstitium (T) of patients with diabetic (DN, G: *n* = 14, T: *n* = 18) and hypertensive (HT, G: *n* = 15, T: *n* = 21) nephropathy and living donors (LD, G: *n* = 42, T: *n* = 42) as controls, revealing a similar expression profile.

B   Heatmap depicting pathway activities in *Foxd1Cre::Pdgfrb+/J* mice and human CKD patients according to PROGENy.

C   Enrichment analyses of nodes in the inferred causal regulatory signaling networks of *Foxd1Cre::Pdgfrb+/J* mice and human CKD patients using the Biocarta database.

D   Based on the gene array data sets, transcription factor activities were estimated by using DoRothEA. Depicted are the most regulated ones regulated both in *Foxd1Cre::Pdgfrb+/J* mice and in human CKD patients.

Source data are available online for this figure.

PROGENy (Schubert *et al*, 2018), a tool to estimate pathway activities by evaluating changes in expression levels of genes affected by perturbation of pathways. PROGENy showed the highest activity in the JAK/STAT pathway, but also high activity in other central pathways known to be active in kidney diseases and fibrosis, like TNFa, NFκB, MAPK, p53, and TGFb. In contrast, VEGF and Trail pathways'

activities were downregulated (Fig 8B). Comparison to the results generated from public microarray data sets of CKD patients (Tajti *et al*, 2019) showed that 9 out of 11 pathways (82%) were similarly regulated in our mice compared to patients. We next generated regulatory causal networks from both murine and human data sets with the tool CARNIVAL (Liu *et al*, 2019). Then, we performed an enrichment analysis using the Biocarta gene set from the MSigDB database on the pathways obtained from CARNIVAL (Fig 8C). This approach also identified nine similarly regulated pathways between our mice and patients with CKD, including MAPK, GSK3, EGF, and, confirmatory to the induced hyperactivation in our mice, the PDGF pathway. In addition, pathways involved in cell cycle control, e.g., p53, ATM, and RACCYCD, and immune response modulation, e.g., IL10, IL22BP, and ETS, were also upregulated in both species. Finally, using DoRothEA, which estimates the activity of transcription factors based on the expression levels of its direct targets, we found that out of the 50 most regulated transcription factors in the *Foxd1Cre::Pdgfrb*$^{+/J}$ mice, 16 were also regulated in the CKD patients' cohort. In particular, Spi1, Stat1, Stat2, Irf1, and Irf2 were strongly upregulated in both mice and humans (Fig 8D).

We also investigated *Sox2Cre::Pdgfrb*$^{+/K}$ mice, which bear a different activating PDGFR-β point mutation (D849V) in the kinase domain (*Pdgfrb*$^{+/K}$) compared to the juxtamembrane mutation in *PDGFRb*$^{+/J}$ mice. In the *Sox2Cre::Pdgfrb*$^{+/K}$ model, hyperactive PDGFR-β-STAT1 signaling mimics a strong interferon signal with connective tissue wasting and tissue-specific infiltration of inflammatory cells and STAT1 deletion significantly prolongs mouse survival (Olson & Soriano, 2011; He *et al*, 2017). The Sox2 promoter is active in all epiblast-derived cells and targets a broad range of cells, including all kidney cells (Hayashi *et al*, 2002). *Sox2Cre::Pdgfrb*$^{+/J}$ and *Sox2Cre::Pdgfrb*$^{+/K}$ mice were phenotypically indistinguishable, with both exhibiting postnatal mortality at approximately 2 weeks after birth (Olson & Soriano, 2011). We analyzed the renal phenotype of young *Sox2Cre::Pdgfrb*$^{+/K}$::*Stat1*$^{+/-}$ mice that survived until 3 weeks of age, and compared them to age-matched *Sox2Cre::Pdgfrb*$^{+/K}$::*Stat1*$^{-/-}$ mice that are rescued from autoinflammation (He *et al*, 2017). At this very early time point, both mouse lines showed a slight increase in mesangial cellularity and mesangial α-SMA staining (Fig EV4), without pathological collagen deposition. While *Sox2Cre::Pdgfrb*$^{+/K}$::*Stat1*$^{+/-}$ mice die at 3 weeks of age, 9-week-old *Sox2Cre::Pdgfrb*$^{+/K}$::*Stat1*$^{-/-}$ mice closely resembled the phenotype of *Foxd1Cre::Pdgfrb*$^{+/J}$ mice, with mesangioproliferative glomerulonephritis, mesangial sclerosis, and interstitial fibrosis (Fig EV4). Thus, despite its high upregulation in the gene array analyses, the profibrotic effects in kidneys are independent of STAT1 and are likely mediated by other PDGFR-β signaling pathways.

### Lack of PDGFR-β in renal mesenchymal cells impairs kidney development

Next, we deleted *Pdgfrb* in renal mesenchymal cells by crossbreeding *Foxd1-Cre* mice and mice with a floxed *Pdgfrb* gene (129S4/SvJae-*Pdgfrb*$^{tm11Sor}$/J—abbreviated as *Foxd1Cre::Pdgfrb*$^{fl/fl}$; Fig EV-5A). Confirmatory, these mice exhibited no glomerular expression of PDGFR-β (Fig EV5B). In contrast to the ubiquitous *Pdgfrb*$^{-/-}$ mice (Soriano, 1994; Lindahl, 1997), *Foxd1Cre::Pdgfrb*$^{fl/fl}$ mice were born at the expected Mendelian ratio (Appendix Table S2). However, prominent developmental renal defects led to death of all

mice at 6–7 weeks after birth. At this time point, kidneys exhibited severe, diffuse, and mostly global glomerular pathology: Ontogenetically youngest glomeruli, located in the outer cortex, showed developmental arrest including small glomeruli and lacked a mesangium and capillary loop formation (Fig EV5C). Older glomeruli in the mid and inner cortex showed segmental and global glomerulosclerosis and segmental rupture of the capillaries with associated extracapillary proliferates, i.e., crescents (Fig EV5C). The presence of mesangial cells in the oldest glomeruli, i.e., those developing most early, suggests that in early development, mesangial cells can be recruited from a *Foxd1*-independent population, potentially from the myelin protein zero (P0) lineage derived from the neural crest (Asada *et al*, 2011). Immunohistochemistry for the mesangial cell marker NG2 confirmed the absence of mesangial cells in developing glomeruli of *Foxd1Cre::Pdgfrb*$^{fl/fl}$ mice (Fig EV5D). Consequently, renal function of *Foxd1Cre::Pdgfrb*$^{fl/fl}$ mice was severely impaired and blood pressure increased (Fig EV5E–G). We did not observe any pathological changes in other organs. Thus, PDGFR-β signaling in mesenchymal cells appears vital for normal glomerulogenesis and kidney development.

### No evidence for tubular cell effects of PDGFR-β expression

Some, but not all, studies described tubular PDGFR-β expression (Alpers *et al*, 1992; Das *et al*, 2017). To investigate the potential role of tubular PDGFR-β signaling, we crossbred *Pdgfrb*$^{+/J}$ mice with mice carrying a tubular specific *Pax8* Cre recombinase (*Pax8Cre::Pdgfrb*$^{+/J}$; Appendix Fig S10A). These mice were born in expected genotype numbers and showed no renal abnormalities (Appendix Fig S10B and Appendix Table S3). The renal function, assessed by BUN (Appendix Fig S10C) and creatinine clearance (Appendix Fig S10D), was not affected.

## Discussion

All currently used animal models analyzing renal fibrosis are induced by injury to non-mesenchymal cells, in particular tubular epithelial cells, glomerular endothelial cells, or podocytes, by toxic, surgical, or (auto)immune mechanisms (Djudjaj & Boor, 2018). In all these preclinical animal models, renal fibrosis develops in response to non-mesenchymal cell injury. This led to the controversial hypothesis that fibrosis might in fact be a protective process similar to wound healing (Hewitson, 2012; Rockey *et al*, 2015; Djudjaj & Boor, 2018). Our mice represent a model of primary, pure renal fibrosis that subsequently drives secondary renal injury, providing direct experimental evidence for the hypothesis that renal fibrosis *per se* is pathological and drives organ failure. Our model enables very specific testing of anti-fibrotic drugs without potential confounding effects of other processes such as inflammation, cell death, and regeneration. In fact, treatment with imatinib showed potent anti-fibrotic effects in our mice, confirming its specificity for targeting fibrosis as a process. This model will also allow us to study very specifically the potential reversibility of renal fibrosis in different renal compartments. Our data with imatinib suggested that while interstitial fibrosis might be partially reversible, glomerular sclerosis is likely not, at least not if the treatment is initiated in advanced and established disease as in our case. This indicates that

glomeruli might have very limited ability to degrade collagen once deposited, whereas in the interstitium, this is partly possible. From a therapeutic view, our data suggest that PDGFR-β inhibition might be most effective if used in an active phase of fibrogenesis characterized by activated mesenchymal cells and not in end-stage, pauci-cellular fibrosis.

While the effects of PDGFR-β activation in mesenchymal cells in the various compartments were similar, i.e., proliferation and profibrotic activation, our model revealed several differences in their consequences. In the glomeruli, prominent and progressive mesangial proliferation preceded progressive sclerosis. These effects, however, did not affect other glomerular cells, i.e., the endothelial cells and the podocytes, and did not result in hematuria or proteinuria. This argues against the current belief that (micro)hematuria can reflect mesangioproliferative changes in the glomeruli. This also suggests that PDGFR-β-driven mesangial cell activation does not induce any pathological cellular cross-talk. This is in contrast to another growth factor, VEGF, which is well known to mediate an intensive cross-talk, in particular between podocytes and glomerular endothelial cells (Eremina et al, 2008).

The sequence of PDGFR-β activation driving glomerular mesenchymal cell proliferation followed by sclerosis appeared reversed in the interstitium, where PDGFR-β activation induced very prominent fibrosis already at the earliest time points studied. This increased only slightly thereafter, whereas activation and expansion of fibroblasts and pericytes was observed later and was progressive. These changes translated into distinct ultrastructural changes of endothelial cells in peritubular capillaries, which are characteristic for chronic kidney disease (Babickova et al, 2017). Our model thereby suggests cellular cross-talk between both cortical fibroblasts and medullary pericytes and endothelial cells for supporting the renal microvasculature. This is in line with previous data in renal pericytes and Gli1[+] cells (Kramann et al, 2015, 2017). The reasons for the differential consequences of activated glomerular versus interstitial mesenchymal cells on endothelial cells are yet unclear, but the highly specialized and unique microvascular structure of glomeruli might be a potential explanation.

Renal anemia complicates chronic kidney diseases in patients. During fibrosis, FoxD1 lineage-derived fibroblasts, all of which express PDGFR-β, were shown to lose their ability to produce EPO as they turn into myofibroblasts (Souma et al, 2013; Chang et al, 2016; Gerl et al, 2016). In agreement with this, our Foxd1Cre::Pdgfrb[+/J] mice also developed progressive anemia due to reduced production of renal EPO and failed to respond to blood loss, mimicking renal anemia in patients. Thus, our study identifies PDGFR-β as a novel pathway involved in the regulation of EPO production.

We have previously shown that a developmental, Sox2-driven constitutive activation of PDGFR-β leads to juvenile lethality due to autoinflammation and lipoatrophy (He et al, 2017). These effects were independent of interferon receptors, but were mediated by STAT1, which regulates genes of the interferon response (He et al, 2017). We also demonstrated previously that PDGFR-β activation drives inflammation and disease progression in murine models of atherosclerosis (He et al, 2015). Interestingly, despite regulation of various transcription factors and pathways involved in inflammation including Stat1 in our mice, we did not find signs of kidney autoinflammation, i.e., vasculitis, tubulitis, or glomerulitis, in Sox2Cre:: Pdgfrb[+/K] or Foxd1Cre::Pdgfrb[+/J] mice, even at advanced age. We

only noted a slight increase in interstitial inflammatory cells, which coincided with the secondary tubular injury. This suggests that the autoinflammatory response regulated by PDGFR-β-STAT1 is tissue-specific and the profibrotic effects in the kidneys are STAT1-independent. Our array data in young Foxd1Cre::Pdgfrb[+/J] mice showed an upregulation of genes involved in fibrogenesis, i.e., cell proliferation, ECM, apoptosis, and autophagy. Interestingly, a set of interferon-response genes was also regulated. This is in line with data showing that targeting interferon specifically to renal fibroblasts effectively reduced renal fibrosis (Boor et al, 2010a; Poosti et al, 2015).

Additional pathways might be involved in the profibrotic phenotype of renal mesenchymal cells in our mice. Several genes involved in autophagy could be detected as regulated in the array, e.g., the tripartite motif (TRIM) proteins Trim5 and its homologues Trim12a, Trim12c, and Trim30 (Mandell et al, 2014; Lascano et al, 2016; Sparrer & Gack, 2018) and cathepsin E (Ctse) (Tsukuba et al, 2013). While nothing is known yet about the role of autophagy specifically in the renal mesenchymal cells, in tubular cells autophagy contributes to renal fibrosis (Huber et al, 2012; Kim et al, 2012; Ding et al, 2014). Other regulated genes detected in the array analysis, in particular the mammalian chitinase-like proteins Chi3l and Chi4l, are extracellular matrix components that were proposed to be involved in extracellular matrix remodeling (Nio et al, 2004). Finally, our gene expression data and computational analyses in diseased human kidneys showed a very similar profile of gene expression and pathway activation compared to our mice with PDGFR-β-induced renal fibrosis, supporting the relevance of these pathways including PDGF signaling for human renal fibrosis. Taken together, our data demonstrate that PDGFR-β activation in renal mesenchymal cells results in a profibrotic gene expression pattern, which primarily drives fibrosis rather than inflammation and is also found in patients with CKD.

Using different transgenic approaches in mice, we found that physiological PDGFR-β signaling in renal mesenchymal cells, but not renal epithelial cells of the Pax8 lineage, is essential for normal kidney development. We used different approaches to manipulate PDGFR-β signaling in mesenchymal cells, including partial and complete deletion of PDGFR-β alleles and an activation using a constitutively active mutant PDGFR-β with a point mutation similar to mutations occurring in human diseases with activating PDGFR-β signaling (Cheung et al, 2013; Martignetti et al, 2013; Johnston et al, 2015; Takenouchi et al, 2015; Arts et al, 2016; Minatogawa et al, 2017). This might have direct relevance for patients with germline activating PDGFR-β mutations, i.e., Kosaki overgrowth and Penttinen syndrome. Currently, these patients have not been analyzed in detail with regard to renal function.

Lack of PDGFR-β in renal mesenchymal cells resulted in defective recruitment of mesangial cells and defective glomerulogenesis. This mirrored previous studies with an germline knockout of PDGFR-β, which was however embryonically lethal, in particular due to cardiovascular malformations (Soriano, 1994). In our study, the mice died at postnatal weeks 5–7, suggesting that the aberrantly developed glomeruli still retained some filtration ability. This allowed us to analyze the postnatal consequences of maldeveloped glomeruli lacking a mesangium, in particular focal and segmental rupture of capillary loops and extracapillary proliferates as well as glomerulosclerosis. These data suggest that glomeruli lacking a mesangium cannot withstand the postnatal functional demands, in particular the filtration pressure, and get progressively damaged,

finally leading to renal failure. Thereby, these data confirm the hypothesis that mesangial cells are essential for glomerular stability (Kriz *et al*, 1995).

In conclusion, our study identifies physiological PDGFR-β signaling in renal mesenchymal cells as essential for normal renal development. Activation of PDGFR-β was sufficient to drive progressive renal fibrosis, and this created a unique model allowing to specifically study the consequences, reversibility, and therapeutic interventions of renal fibrosis independent of inflammation, hypertension, or epithelial or endothelial injury.

# Materials and Methods

### Human tissue

Human kidney samples from nephrectomy specimens from the Institute of Pathology and the centralized biomaterial bank were processed in a retrospective and anonymous manner. Informed consent was obtained from all subjects processed by the centralized biobank. The study was approved by the local review board of Aachen in line with the WMA Declaration of Helsinki and the Department of Health and Human Services Belmont Report.

### Animals

Animal experiments were performed in accordance with Guide for the Care and Use of Laboratory Animals and were approved by the local review boards and authorities. Mouse strains FoxD1$^{GC}$ (012463), Pax8Cre (028196), tdTomato reporter mice (B6;129S6-*Gt(ROSA) 26Sor*$^{tm9(CAG-tdTomato)Hze}$/J) (007905), 129S4/SvJae-*Pdgfrb*$^{tm11Sor}$/J (017622), Sox2-Cre (008454), and STAT1$^{flox}$ (012901) are available at the Jackson Laboratory. *Pdgfrb*-eGFP mice (Tg(*Pdgfrb-EGFP*) JN169Gsat/Mmucd) are available at Mutant Mouse Resource and Research Center (031796-UCD). Generation of Pdgfrb$^{(S)J}$ and Pdgfrb$^{(S)K}$ is described in a previous publication (Olson & Soriano, 2011).

Mice were examined in a mixed C57BL/6;129Sv genetic background. If not specified, both male and female mice were used in the experiments. All animal experiments were approved by the local

review boards. Animals were held under SPF conditions in filtertop cages in rooms with constant temperature and humidity and 12-h/12-h light cycles and had *ad libitum* access to drinking water and food.

For characterization of *Foxd1Cre::Pdgfrb*$^{+/J}$ mice in a time course, mice 6, 14, 25, and 35 weeks of age were examined. *Foxd1Cre:: Pdgfrb*$^{fl/fl}$ mice were examined at an age of 1 and 6 weeks.

For genotyping, DNA from tail biopsies was isolated with the Qiagen DNeasy® Blood & Tissue Kit (Qiagen). The primers are listed in Table 1. PCR products were separated by electrophoresis on a 2% agarose/TBE gel containing GelRed (Biotium).

### Functional parameter analyses

Blood pressure was measured via the tail-cuff method (CODA®; Kent Scientific). Data show values for systolic blood pressure.

For urine collection, mice were held in metabolic cages (Tecniplast) for 15 h with *ad libitum* drinking water.

Blood was taken retro-orbital with heparinated glass canulaes and collected in EDTA tubes for blood picture or in serum tubes (Sarstedt). Serum was separated by centrifugation at 1,150 $g$ at 4°C for 10 min. Blood urea nitrogen, serum creatinine, and urinary creatinine were measured with automatized colorimetric detection using an autoanalyzer.

Serum EPO concentrations were measured with a solid-phase sandwich mouse EPO Quantikine ELISA kit (R&D Systems). The assay was performed according to the manufacturer's manual.

### Anemia induction

Anemia was induced in 13-week-old mice by taking off 200 μl blood retro-orbital under isoflurane anesthesia. The same amount of 0.9% NaCl solution was given intraperitoneally to compensate the volume loss. This procedure was repeated after 24 h. After a recovery time of 5 days, mice were sacrificed and blood was taken for final analysis.

### Unilateral ureter obstruction model

Ten-week-old *Foxd1Cre::Pdgfrb*$^{+/J}$ ($n = 3$) and wt littermates ($n = 4$) as well as eGFP reporter mice underwent unilateral ureter

**Table 1. Sequences of genotyping primers.**

| Gene | Primer | Sequence | Band size | Annealing temp. |
|------|--------|----------|-----------|-----------------|
| *FoxD1Cre* | Common | TCT GGT CCA AGA ATC CGA AG | – | 58°C |
| | Wild type | CTC CTC CGT GTC CTC GTC | 250 bp | |
| | Mutant | GGG AGG ATT GGG AAG ACA AT | 450 bp | |
| *Pax8Cre* | Forward | TGT CCC TGA CAA TTT GGT CTG TT | | 58°C |
| | Reverse | GAT CCT GGC AAT TTC GGC TAT AC | 600 bp | |
| *Pdgfrb*$^{(S)J}$ | Common | GGG CTT CCA GGA GTG ATA CC | – | 60°C |
| | Wild type | CCA GCT GGA CTG AAG AGG AG | 346 bp | |
| | Mutant | CCG AGC AGG TCA GAA CAA AG | 160 bp | |
| *Pdgfrb*$^{(S)J}$ flox efficacy | Forward | CAA ACT CTT CGC GGT CTT TC | 200 bp | 56°C |
| | Reverse | CTG GTA TCA CTC CTG GAA GC | – | |
| *Pdgfrb flox* | Forward | CCA GTT AGT CCA CTT ATG TTG | 389 bp (wt) | 50°C |
| | Reverse | TAC CAG GAA GGC TTG GGA AG | 520 bp (flox) | |

obstruction (UUO) by cauterization of the ureter, as described previously (Buhl *et al*, 2016). Mice were sacrificed 5 days after UUO, and the kidneys were harvested.

## Angiotensin II infusion

Angiotensin II (Sigma) was solved in sterile 0.9% NaCl solution and administered *via* osmotic pumps (ALZET® 1004) at a dose of 1,000 ng/kg/min, according to a common approved protocol (Lu *et al*, 2015). Pumps were implanted subcutaneously under isoflurane anesthesia in 10-week-old male mice. Blood pressure measurements and serum and urine collection were done before pump implantation and before sacrifice.

## Imatinib intervention

Imatinib mesylate (Gleevec®; Novartis) was dissolved in water and administered daily orally *via* gavage at a dose of 50 mg/kg bodyweight over a period of 3 weeks in 22-week-old mice using a common dosing protocol (Sadanaga *et al*, 2005; Zoja *et al*, 2006; Iyoda *et al*, 2009). Control animals got pure water.

## Histology, immunohistochemistry, and immunofluorescence

Tissue for light microscopy and immunohistochemistry was fixed in methyl Carnoy's solution, dehydrated, embedded in paraffin, and cut into 1-μm sections. Sections were stained with periodic acid–Schiff's reagent and counterstained with hematoxylin.

Methenamine silver staining (Gömöri) was performed according to the standard protocol as described recently (Evers *et al*, 2016).

Immunohistochemistry was performed as described previously (Boor *et al*, 2007, 2010b; Buhl *et al*, 2016; Ehling *et al*, 2016). The primary and secondary antibodies used in this study are listed in Table 2. Nuclei were counterstained with methyl green. For negative controls, substitution of PBS buffer or irrelevant matched IgG in place of the primary antibodies was used; no unspecific staining was observed in any of the stainings.

Tissue from tdTomato or eGFP reporter mice was fixed in 4% PFA, incubated in 30% sucrose solution overnight, and embedded in OCT (Sakura Finetek). 7-μm sections were cut on a cryotome (Leica).

Light and immunofluorescent stainings were analyzed using a Keyence BZ-9000 microscope.

## Morphometry

For morphometric analyses, stained sections were scanned using a whole-slide scanner (NanoZoomer HT; Hamamatsu Photonics), and imaging software (NDP.view and ImageJ; National Institutes of Health) was used as follows: For each slide, 8–16 images in ×40 magnification were obtained and the percentage of positively stained area of each slide was quantified. All analyses and quantifications were performed in a blinded manner.

## Tissue clearing

Optical tissue clearing was performed as described previously (Puelles *et al*, 2019). Thick slices were physically cut mid-hilar with an in-house-made device and then fixed with 4% PFA. After 24 h of fixation, slices were washed in PBS overnight and then incubated in Sca*l*e media (ScaleView-A2; Wako Chemicals, Cat#: 193-18455) for 2 weeks. 3D stacks were acquired using a two-photon microscope (Olympus FV1000MPE; Olympus Corp., Tokyo, Japan) fitted with 25× water immersion objective lens (NA: 1.05) with correction collar for refractive index adjustment attached to a pulsed Ti:sapphire laser (MaiTai DeepSee; Spectra Physics, Mountain View, USA). 3D rendering of a 300-μm stack was conducted with Imaris v9.1 (Bitplane) software.

## Transmission electron microscopy

Tissue was fixed in Karnovsky fixative. Post-fixation of cells was performed in 1% $OsO_4$. After fixation, samples were dehydrated by an ethanol series (30, 50, 70, 90, and 100%). The dehydrated specimens were incubated in propylene oxide (Serva) and embedded in Epon. Ultrathin sections (70–100 nm) were stained with uranyl acetate and lead citrate (all EMS). The specimens were viewed using a Zeiss LEO 906 electron microscope, operated at an acceleration voltage of 60 kV.

## RNA *in situ* hybridization

RNA *in situ* hybridization was performed on formalin-fixed paraffin sections using the RNAscope® Multiplex Fluorescent Reagent Kit v2 (Advanced Cell Diagnostics) according to the manufacturer's instructions. The probes used were targeted to murine *Epo* (Cat No. 315501) and murine *Pdgfrb* (Cat No. 411381).

## RNA extraction and analysis

Total RNA was extracted from renal cortex from mice using the RNeasy Mini Kit (Qiagen). Complementary DNA (cDNA) syntheses and real-time quantitative PCRs were performed as previously described (Eitner *et al*, 2008; Boor *et al*, 2010b). All quantitative data from the RT–qPCR were normalized to GAPDH (glyceraldehyde 3-phosphate dehydrogenase). Sequences of primers and probes are given in Table 3.

## Western blotting

For Western blot analyses, equal amounts of cortex protein lysates were separated in 4–12% Bis-Tris gradient gel under reducing conditions. Proteins were transferred to nitrocellulose membranes. Nonspecific binding sites were blocked in Tris-buffered saline containing 5% (w/v) non-fat milk powder. Primary antibodies (Table 2) were visualized using horseradish peroxidase-conjugated IgG (Table 2) and SuperSignal chemiluminescent substrate (Pierce).

## Cell isolation

Primary cell isolation was performed as previously described (Djudjaj *et al*, 2016). Mice were perfused with PBS containing ferric oxide particles (50–100 nm, 18 mg/ml; Thermo Fisher). Kidneys were extracted, minced, and incubated in 1 mg/ml collagenase IV (Worthington). For mesangial cell outgrowth, glomeruli were separated *via* magnetic separation and transferred in RPMI media supplemented with 10% fetal calf serum, 2 mM L-glutamine, 100 U/ml penicillin, and 100 U/ml streptomycin. Fibroblasts

**Table 2.  Antibodies used in immunohistochemistry and Western blot.**

| Target | Clone/Cat. No. | Host | Application | Dilution | Supplier |
|---|---|---|---|---|---|
| CD45 | 30-F11 | Rat | IHC | 1:200 | BD Pharmingen |
| Collagen type I | 1310-01 | Goat | IHC | 1:100 | Southern Biotech |
| Collagen type III | 1330-01 | Goat | IHC | 1:100 | Southern Biotech |
| ErHr3 | T-2012 | Rat | IHC | 1:100 | BMA Biomedicals |
| F4/80 | CI:A3-1 | Rat | IHC | 1:800 | Serotec |
| Fibronectin | AB1954 | Rabbit | IHC | 1:500 | Millipore/Merck |
| Ki67 | TEC-3 | Rat | IF | 1:50 | Dako/Agilent |
| Lipocalin-2 | AF1857 | Goat | IHC | 1:450 | R&D system |
| NG-2 | 546930 | Rat | IHC | 1:100 | R&D system |
| PDGFR-β | Y92 | Rabbit | IF | 1:100 | Abcam |
| Renin | ABIN238050 | Sheep | IHC | 1:10.000 | Antibodies-online |
| α-SMA | 1A4 | Mouse | IHC | 1:500 | Dako/Agilent |
| Goat IgG | Biotin | Horse | IHC | 1:300 | Vector Laboratories |
| Mouse IgG | Biotin | Horse | IHC | 1:300 | Vector Laboratories |
| Rabbit IgG | Biotin | Goat | IHC | 1:300 | Vector Laboratories |
| Rabbit IgG | Alexa-647 | Donkey | IF | 1:200 | Jackson Immuno Research |
| Rat IgG | Biotin | Goat | IHC | 1:300 | Vector Laboratories |
| Rat IgG | Alexa-488 | Donkey | IF | 1:200 | Life Technology |
| Sheep IgG | Biotin | Rabbit | IHC | 1:300 | Vector Laboratories |
| β-Actin | A5441 | Mouse | WB | 1:1,000 | Sigma-Aldrich |
| PDGFR-β | 28E1 #3169 or sc-432 | Rabbit | WB | 1:1,000 | Cell Signaling Santa Cruz |
| p-PDGFR-β | Y1009 42F9 #3124 | Rabbit | WB | 1:1,000 | Cell Signaling |
| Akt | 11E7 #4685 | Rabbit | WB | 1:1,000 | Cell Signaling |
| p-Akt | S473 D9E#4060 | Rabbit | WB | 1:1,000 | Cell Signaling |
| ERK1/2 | #9102 | Rabbit | WB | 1:1,000 | Cell Signaling |
| p-ERK1/2 | T202/Y204 #9101 | Rabbit | WB | 1:1,000 | Cell Signaling |
| PLCγ | B.-6-4 | Mouse | WB | 1:1,000 | Millipore/Merck |
| p-PLCγ | Y783 #2821 | Rabbit | WB | 1:1,000 | Cell Signaling |
| JNK | 56G8 #9258 | Rabbit | WB | 1:1,000 | Cell Signaling |
| p-JNK | T183/Y185 81E11 #4668 | Rabbit | WB | 1:1,000 | Cell Signaling |
| p38 | T180/Y182#612281 | Mouse | WB | 1:1,000 | BD Biosciences |
| p-p38 | #612168 | Mouse | WB | 1:1,000 | BD Biosciences |
| A-Tubulin | B-7 sc-5286 | Mouse | WB | 1:1,000 | Santa Cruz |
| Mouse Ig | HRP, sc-2005 | Goat | WB | 1:10,000 | Santa Cruz |
| Rabbit Ig | HRP, sc-2004 | Goat | WB | 1:10,000 | Santa Cruz |

IF, immunofluorescence; IHC, immunohistochemistry; WB, Western blot.

were cultivated in DMEM supplemented with 10% fetal calf serum, 2 mM L-glutamine, 100 U/ml penicillin, and 100 U/ml streptomycin. Cells were tested for mycoplasma infection with MycoTool (Roche) mycoplasma detection kit and were found to be not infected.

**BrdU proliferation assay**

Cell proliferation was determined by 5-bromo-2′-deoxyuridine (BrdU) incorporation and detection with a colorimetric BrdU ELISA (Roche)

according to the manufacturer's manual. Cells of the $3^{rd}$–$4^{th}$ passage were used for the assay. BrdU incorporation was allowed for 24 h.

**Microarray of murine tissue**

For microarray analysis, renal cortex tissue of three *Foxd1Cre:: Pdgfrb*$^{+/J}$ and three wt littermates 6 weeks of age was used. RNA was isolated as described above. RNA quality and quantity were assessed using NanoDrop (Thermo Fisher Scientific) and Agilent Bioanalyzer (Agilent Technologies). For microarray analysis,

100 ng RNA per sample was transcribed and labeled with Gene-Chip WT PLUS Reagent Kit (Affymetrix) according to the manu-facturer's protocol. Labeled cDNA was hybridized to Affymetrix GeneChip Mouse Transcriptome Array 1.0 and visualized with the Affymetrix staining kit. The Affymetrix Expression Console soft-ware was utilized for gene data analysis. Data were normalized with the SST-RMA algorithm. Robust mean signal values were calculated using the Tukey biweight estimator. For calculating significance ($P$-value), the non-parametric Wilcoxon rank test was used. Probe sets with a $P$-value < 0.05 were considered as dif-ferentially expressed.

**Microarray analysis of human kidney biopsies**

Human renal biopsy specimens and Affymetrix microarray expres-sion data were obtained within the framework of the European Renal cDNA Bank–Kröner-Fresenius Biopsy Bank (Cohen *et al*, 2002). Biopsies were obtained from patients after informed consent and with approval of the local ethics committees. Following renal biopsy, the tissue was transferred to RNase inhibitor and microdissected into glomeruli and tubulointerstitium. Total RNA was isolated from microdissected glomeruli, reverse-transcribed, and linearly amplified according to a protocol previously reported (Cohen *et al*, 2006). Affymetrix GeneChip Human Genome U133A and U133 Plus2.0 Arrays were used. In this study, we used the published microarray expression data from individual patients with diabetic nephropathy and hypertensive nephropathy (Data ref: Shved *et al*, 2017a), and living donors (Data ref: Berthier *et al*, 2012a,b; Data ref: Reich *et al*, 2010). Pre-transplantation kidney biopsies from living donors were used as control. Raw data were normalized using Robust Multichip Algorithm (RMA) and annotated by Human Entrez Gene custom CDF annotation version 18 (http://brainarray.mbni.med.umich.edu/ Brainarray/default.asp). The log-transformed data set was corrected for batch effect using ComBat from the GenePattern pipeline (http:// www.broadinstitute.org/cancer/software/genepattern/). To identify differentially expressed genes, the SAM (Significance Analysis of Microarrays) method was applied using TiGR (MeV, version 4.8.1) (Tusher *et al*, 2001). A $q$-value below 5% was considered to be statistically significant.

**Computational functional analyses of microarray data**

The fold changes in gene expression of *Foxd1Cre::Pdgfrb*$^{+/J}$ mice compared to controls were applied to estimate the activities of regu-latory transcriptional factors and of signaling pathway using DoRothEA (Garcia-Alonso *et al*, 2018) and PROGENy (Schubert

*et al*, 2018), respectively. The activities of top 50 regulatory tran-scription factors and signaling pathways were integrated into a signed-directed human protein–protein interaction network obtained from OmniPath (Turei *et al*, 2016) to infer upstream regulatory network structures using CARNIVAL (Liu *et al*, 2019). Over-repre-sentation analyses of the nodes that were predicted to be upregu-lated were performed using the Biocarta gene sets in the curated (C2) branch from MSigDB (Subramanian *et al*, 2005). $P$-values were represented in the minus $\log_{10}$ scale, and the significance of enrich-ment results was determined with $P$-value < 0.05. The same analyti-cal pipeline was applied to the human CKDs public microarray data sets compared to the ones of healthy living donors as detailed in Tajti *et al* (2019).

**The paper explained**

**Problem**
Chronic kidney disease (CKD) affects around 10% of the world popula-tion and is associated with substantial mortality and morbidity. The underlying disease process and common final pathway of CKD is kidney scarring (or fibrosis), i.e., pathological deposition of connective tissue, i.e., extracellular matrix, and loss of functional kidney tissue. Currently, there are no specific treatment options to treat renal scar-ring, in part due to the lack of fibrosis-specific models.

**Results**
We here show that a common feature of kidney scarring in patients is increased expression and activation of the platelet-derived growth factor receptor (PDGFR)-β in mesenchymal cells. To address the rele-vance of this finding, we have developed a mouse model specifically mimicking this mechanism, i.e., mice with constitutive activation of PDGFR-β in kidney mesenchymal cells. These mice showed specific activation of these cells and kidney scarring, reflecting various aspects found in CKD in patients. Importantly, the scarring in these mice was primary, in contrast to all other currently available models, which develop fibrosis secondary to injury to other cell types, i.e., epithelial, endothelial, or inflammatory. At later time points, primary kidney scarring led to secondary epithelial injury and inflammation, reduced renal function, and anemia. These mice also exhibited an aggravated course of various kidney disease models, suggesting that PDGFR-β activation can aggravate kidney diseases.

**Impact**
Taken together, our data bring novel insights into the pathogenesis of kidney fibrosis, suggesting PDGFR-β signaling as a potential treatment target in fibrosis. Our model specifically proves the detrimental conse-quences of fibrosis and provides a framework for specific testing of anti-fibrotic drugs.

**Table 3. Sequences of primers and probes for RT–qPCR.**

| Gene | Forward primer | Reverse primer |
|---|---|---|
| *Lcn2* | GGC CTC AAG GAC GAC AAC A | TCA CCA CCC ATT CAG TTG TCA |
| *Ifit1* | GCA TCA CCT TCC TCT GGC TAC | AGC CAT GCA AAC ATA GGC CA |
| *Irf8* | GGC AGT GGC TGA TCG AAC A | GGT CTT CTC ATC ATT TTC CCA GA |
| *Casp1* | ACT GACT GGG ACC CTC AAG T | GCA AGA CGT GTA CGA GTG GT |
| *Trim30d* | GTC ATG GAA ATG AAG CAG GGT G | TTG GGG GAA CTA TTT GGG GC |
| *Gapdh* | GGC AAA TTC AAC GGC ACA GT | AGA TGG TGA TGG GCT TCC C |

## Statistical analysis

Data are presented as mean ± SD. Means between the groups were compared using two-tailed unpaired Student's *t*-test using GraphPad Prism version 6.01 (GraphPad Software Inc.), and outliers were identified by the ROUT method. *P*-values < 0.05 were considered significant. Exact *P*-values are stated in Appendix Table S4.

# Data availability

The microarray data from murine array have been deposited to the GEO database (https://www.ncbi.nlm.nih.gov/geo/) and assigned the identifier GSE138819.

**Expanded View** for this article is available online.

## Acknowledgements

We acknowledge C. Timm, C. Gianussis, and S. Otten for assistance with histopathology. This study was financed by the German Research Foundation (DFG: SFB/TRR57 and SFB/TRR219, BO3755/3-1, and BO3755/6-1), the German Ministry of Education and Research (BMBF: STOP-FSGS-01GM1901A), and the RWTH Interdisciplinary Centre for Clinical Research (IZKF: O3-7). VGP was supported by the National Health and Medical Research Council (NHMRC) of Australia (CJ Martin Research Fellowship; 1128582), the Humboldt Foundation, and the German Society of Nephrology (DGfN). TBH was supported by the DFG (CRC 1192, CRC1140, CRC 992, HU 1016/8-2), by the BMBF (STOP-FSGS 01GM1901A), by the European Research Council (ERC Grant 61689, DNCure), and by the H2020-IMI2 consortium BEAt-DKD (Innovative Medicines Initiative 2 Joint Undertaking under Grant Agreement No. 115974). The ERCB-KFB was supported by the Else Kröner-Fresenius Foundation. PT is supported by the Innovative Medicine Initiative 2 Joint Undertaking TransQST (116030) to JSR. This Joint Undertaking receives support from the European Union's Horizon 2020 research and innovation programme and EFPIA. We also thank all participating centers of the European Renal cDNA Bank–Kröner-Fresenius Biopsy Bank (ERCB-KFB) and their patients for their cooperation. For active members at the time of the study, see Shved *et al* (2017b).

## Author contributions

EMB and PB were responsible for the overall study design. JF and RW gave scientific advice. EMB, SD, BMK, KE, VGP, and EB-K performed and analyzed the experiments. CH and LEO collected and provided kidney material of *Sox2Cre::Pdgfrb$^{+/K}$::Stat1$^{-/-}$* mice. BD performed and analyzed the murine microarray. PT and JS-R performed computational functional analyses of murine microarray data. MTL, CDC, and TBH collected, performed, and analyzed the human microarray. EMB and PB wrote the manuscript. All authors critically reviewed, discussed, and commented on the manuscript.

## Conflict of interest

The authors declare that they have no conflict of interest.

## For more information

Authors' webpage: https://www.LaBooratory.ukaachen.de
KIT (Kidney Interactive Transcriptomics): http://humphreyslab.com/SingleCell

Human Entrez Gene custom CDF annotation version 18: http://brainarray.mbni.med.umich.edu
GenePattern pipeline: http://www.broadinstitute.org/cancer/software/genepattern
GEO database: https://www.ncbi.nlm.nih.gov/geo/

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
