## [Review Process File · EMBO Molecular Medicine]

Dysregulated mesenchymal PDGFR- β drives kidney fibrosis

Eva M. Buhl, Sonja Djudjaj, Barbara M. Klinkhammer, Katja Ermert, Victor G. Puelles, Maja T. Lindenmeyer, Clemens D. Cohen, Chaoyong He, Erawan Borkham-Kamphorst, Ralf Weiskirchen, Bernd Denecke, Panuwat Trairatphisan, Julio Saez-Rodriguez, Tobias B. Huber, Lorin E. Olson, Jürgen Floege, Peter Boor

Review timeline:

Submission date:	3 July 2019
Editorial Decision:	29 July 2019
Revision received:	23 October 2019
Editorial Decision:	7 November 2019
Revision received:	8 December 2019
Accepted:	9 December 2019

Editor: Lise Roth

Transaction Report:

1st Editorial Decision

29 July 2019

Thank you for the submission of your manuscript to EMBO Molecular Medicine, and please accept my apologies for the delay in replying due to my recent traveling. We have now received feedback from the three reviewers who agreed to evaluate your manuscript. As you will see from the reports below, the referees acknowledge the interest of the study and are overall supporting publication of your work pending appropriate revisions.

Addressing the reviewers' concerns in full will be necessary for further considering the manuscript in our journal, and acceptance of the manuscript will entail a second round of review. EMBO Molecular Medicine encourages a single round of revision only and therefore, acceptance or rejection of the manuscript will depend on the completeness of your responses included in the next, final version of the manuscript. For this reason, and to save you from any frustrations in the end, I would strongly advise against returning an incomplete revision.

I look forward to receiving your revised manuscript.

***** Reviewer's comments *****

Referee #1 (Remarks for Author):

Buhl et al demonstrated that constitutively active form of PDGFR- β signalling (Foxd1Cre::Pdgfrb+/J::tdTomato mice) drives development and progression of kidney fibrosis in different mouse disease models. They used sophisticated mouse models and techniques to provide direct evidence that fibrosis per se can drive chronic organ damage and establish a model of primary fibrosis allowing specific studies targeting fibrosis progression and regression. This is the revised manuscript. I have following questions.

(1) Figure 1 title: PDGFR- β expression is upregulated in fibrotic human kidneys. It actually includes mouse kidneys.

- (2) Figure 2, they need to show the evidence of activation of downstream of PDGFR- β signalling. They also need to use mesenchymal marker+/Ki67+ to calculate proliferation of mesenchymal cells in vivo.
- (3) Have they quantified time-course inflammation? Inflammation usually precedes fibrosis or co-exists. If fibrogenesis precedes inflammation, it would strengthen their hypothesis that fibrosis per se can drive chronic organ damage.
- (4) Figure 6, they need to use mesenchymal marker(+)/EPO(+) to show the origin of EPO.
- (5) Figures 10 and 11 give confusing information.

Referee #2 (Remarks for Author):

Buhl et al report an original approach and new model to explore the role and contribution of fibrosis itself in CKD progression. This is clinically relevant since fibrosis-targeting drugs are dismissed by critics that argue that they target a late process that is secondary to kidney injury rather than causative. A wide arrange of methods and models have been used to characterize the role of PDGFRbeta and set u the fibrosis-induced CKD model.

I have only minor comments

1. figure 5. Ccr and BUN data do not match well. Ccr here and in fig 8 d, e suggests early hyperfiltration that is surprisingly associated to higher BUN in pdgfrb overactivity mice. Do the authors have serum creatinine data? is there any reason for this apparent discrepancy? Did all mice groups eat the same amount?
2. fig 7. what was the impact of imatinib on renal function?
3. fig 9. please provide full array results as suppl data

Referee #3 (Remarks for Author):

In this manuscript, Buhl and colleagues evaluated the role of Pdgfr-beta in kidney mesenchymal cells. They show that activation of the receptor is enough to drive kidney fibrosis. Importantly pharmacological inhibition of the receptor by imatinib could reverse fibrosis in the tubulointerstitium but not in the glomeruli. This model will be very useful to test new experimental treatments, which are much needed for this disease.

The study also unravels a key role for mesenchymal (but not tubular) Pdgfrb expression in kidney development. Although the data is not essential to the paper (and not mentioned in the abstract), it is a nice addition.

Overall, the manuscript includes a large amount of novel, interesting high-quality data, profoundly changing our understanding of the role of Pdgfrb in kidney development and fibrosis.

Minor details:

- The amino-acid changes corresponding to "J" and "K" mutations should be mentioned.
- In figure 1B, there is confusion between panels (e) and (a) in the legend.
- In figure 5A, please show the data points.
- In figure 10, the end of the caption "and suggested a role for STAT1" is confusing, since the text concludes that the process is STAT1-independent.
- Supplementary figure 12: please correct the spelling of "Makrophages"

Referee #1 (Remarks for Author):

- (1) Figure 1 title: PDGFR- β expression is upregulated in fibrotic human kidneys. It actually includes mouse kidneys.

Authors' reply: Thanks for pointing this out, we have now corrected this ("..in fibrotic human and murine kidneys").

(2) Figure 2, they need to show the evidence of activation of downstream of PDGFR- β signalling. They also need to use mesenchymal marker⁺/Ki67⁺ to calculate proliferation of mesenchymal cells in vivo.

Authors' reply: As suggested, we have performed WB analyses, showing activation of pathways downstream of PDGFR β , particularly of Akt and p38 (added as Figure EV2 in manuscript and show here).

Unfortunately, even after multiple attempts using various approaches, we were not able to establish Ki67 co-staining with mesenchymal markers of sufficient specificity, which would allow rigorous enumeration.

We hope that our data on FoxD1-Tomato⁺ cells, which specifically mark the interstitial mesenchymal cells and show clear expansion of these cells in the mutant mice (Figure 2 F and G), and our in vitro data on proliferation of primary isolated mesangial cells and fibroblasts (Figure 2 H) are convincing enough to prove mesenchymal proliferation.

(3) Have they quantified time-course inflammation? Inflammation usually precedes fibrosis or co-exists. If fibrogenesis precedes inflammation, it would strengthen their hypothesis that fibrosis per se can drive chronic organ damage.

Authors' reply: We analyzed inflammation in the time-course in both glomeruli and interstitium (see Figures 4, Supp. Fig. 4 and 6). These data showed only a slight increase in influx of inflammatory cells (microinflammation at best), which was only observed at late time-points, coincident with tubular injury. On the contrary, fibrosis was already obvious at earliest analyzed time-point, i.e. week 6. Similarly to the reviewer, we also believe that this strengthens our hypothesis that fibrosis per se can drive organ damage.

(4) Figure 6, they need to use mesenchymal marker(+)/EPO(+) to show the origin of EPO.

Authors' reply: We now used Pdgfrb and Epo ISH to further confirm that EPO is produced by renal fibroblasts, well in line with the current dogma (see Figure 6E in the manuscript and below)

(5) Figures 10 and 11 give confusing information.

Authors' reply: Figure 10 shows the data on the experiments using an additional mouse line, with another promotor (Sox2) and another PDGFRb mutation (D849V), i.e. the Sox2Cre::Pdgfrb^{+K} mice. This line exhibits a similar phenotype as Foxd1Cre::Pdgfrb^{+/-} mice, supporting our findings in two independent mice models. The figure also shows data on the role of Stat1 by using Sox2Cre::Pdgfrb^{+K}::Stat1^{-/-} mice. The data shown in this figure are discussed on page 15. We now also rephrased the figure legend.

Figure 11 shows the effect of Pdgfrb deletion in Foxd1Cre::Pdgfrb^{fl/fl} mice, which resulted in failed glomerulogenesis and associated early postnatal death due to kidney failure. These data confirm the essential role of Pdgfrb signaling in normal embryonic development.

The data shown in this figure are discussed on page 16.

We hope that now, together with the discussion of the data, the information of the figures is less confusing.

Referee #2 (Remarks for Author):

I have only minor comments

1.figure 5. Ccr and BUN data do not match well. Ccr here and in fig 8 d, e suggests early hyperfiltration that is surprisingly associated to higher BUN in pdgfrb overactivity mice. Do the authors have serum creatinine data? Is there any reason for this apparent discrepancy? Did all mice groups eat the same amount?

Authors' reply: The data on differences in creatinine clearance at early time-points might seem somewhat higher in the mutant mice, but were not statistically significant. Similarly, the serum creatinine values were not significantly different between the groups at this age (see figure below). At the same time the urinary volume was somewhat smaller in the Foxd1Cre::Pdgfrb^{+/-} mice (not statistically significant). We therefore do not think that these mice might have an early hyperfiltration.

Unfortunately, we have not measured the food intake in these mice.

2. fig 7. what was the impact of imatinib on renal function?

Authors' reply: We haven't seen any significant impact on renal function (see figure below), most likely due to relatively short treatment duration combined with rather mild renal function decline in the *Foxd1Cre::Pdgfrb^{+J}* at this age.

3. fig 9. please provide full array results as suppl data

Authors' reply: To comply with the reviewer's suggestion and the journal regulations, we have submitted our data to the GEO repository. The reference to this database is now also included in the data availability section (Page 31).

Referee #3 (Remarks for Author):

Minor details:

The amino-acid changes corresponding to "J" and "K" mutations should be mentioned.

Authors' reply: PDGFR β J mutant isoform has a V536A mutation in the juxtamembrane domain, whereas the PDGFR β K mutant isoform is a D849V mutation, which is located to the kinase domain. Both amino acid changes are now mentioned in the main text on page 7 and 15:

Page 7: "To generate mice with a renal mesenchymal cell-specific, constitutively active PDGFR- β , we used mice in which one wt *Pdgfrb* allele was substituted by a conditional knock-in of *Pdgfrb* with an activating point mutation (V536A) in the juxtamembrane domain of PDGFR- β , denoted as "J" (*Pdgfrb^{+J}*), behind a floxed STOP cassette 33."

Page 15: "We also investigated *Sox2Cre::Pdgfrb^{+K}* mice, which bear a different activating PDGFR- β point mutation (D849V) in the kinase domain (*Pdgfrb^{+K}*) compared to the juxtamembrane mutation in *PDGFR^{+J}* mice."

In figure 1B, there is confusion between panels (e) and (a) in the legend.

Authors' reply: We are sorry for this, now corrected.

In figure 5A, please show the data points.

Authors' reply: As requested, we now show the data points in this figure.

In figure 10, the end of the caption "and suggested a role for STAT1" is confusing, since the text concludes that the process is STAT1-independent.

Authors' reply: Thanks, we have now corrected and rephrased it:

Figure 10: "Glomerular pathology in Sox2Cre::Pdgfrb+/K mice resembled the findings in Foxd1Cre::Pdgfrb+/J but suggested no role for STAT1"

Supplementary figure 12: please correct the spelling of "Makrophages"

Authors' reply: Thanks for pointing this out, now corrected.

2nd Editorial Decision

7 November 2019

Thank you for the submission of your revised manuscript to EMBO Molecular Medicine. We have received the referees' reports, and as you will see they are now supportive of publication of your study. I am therefore pleased to inform you that we will be able to accept your manuscript pending minor editorial amendments.

I look forward to reading a new revised version of your manuscript as soon as possible.

***** Reviewer's comments *****

Referee #1 (Comments on Novelty/Model System for Author):

No more questions.

Referee #1 (Remarks for Author):

It is an original research manuscript but it is not a research report in that it doesn't contain a specific finding, model or methodology with a high impact on the field of molecular medicine.

Referee #2 (Remarks for Author):

My concerns have been addressed

2nd Revision - authors' response

8 December 2019

Authors made the requested editorial changes.

Corresponding Author Name: Peter Boor

Manuscript Number: EMM-2019-11021